# THE GAUSSIAN-HEAD OFL FAMILY: ONE-SHOT FEDERATED LEARNING FROM CLIENT GLOBAL STATISTICS

**Fabio Turazza**[1][*]**, Marco Picone**[1,2]**, Marco Mamei**[1,2]
[1]Department of Sciences and Methods for Engineering
[2]Artificial Intelligence Research and Innovation Center
University of Modena and Reggio Emilia, Reggio Emilia, Italy
`name.surname@unimore.it`

## ABSTRACT

Classical Federated Learning relies on a multi-round iterative process of model exchange and aggregation between server and clients, with high communication costs and privacy risks from repeated model transmissions. In contrast, one-shot federated learning (OFL) alleviates these limitations by reducing communication to a single round, thereby lowering overhead and enhancing practical deployability. Nevertheless, most existing one-shot approaches remain either impractical or constrained, for example, they often depend on the availability of a public dataset, assume homogeneous client models or require uploading additional data or model information. To overcome these issues, we introduce the **G**aussian-**H**ead **OFL** (GH-OFL) family, a suite of one-shot federated methods that assume class-conditional Gaussianity of pretrained embeddings. Clients transmit only sufficient statistics (per-class counts and first/second-order moments) and the server builds heads via three components: **(i)** Closed-form Gaussian heads (NB/LDA/QDA) computed directly from the received statistics; **(ii)** FisherMix, a linear head trained on synthetic samples drawn in an estimated Fisher subspace; and **(iii)** Proto-Hyper, a lightweight low-rank residual head that refines Gaussian logits via knowledge distillation on those synthetic samples. In our experiments, GH-OFL methods deliver state-of-the-art robustness and accuracy under strong non-IID skew while remaining strictly data-free.

## 1 INTRODUCTION

Federated Learning (FL) enables distributed training without centralizing raw data: clients (e.g., phones, IoT, edge) keep their data local and share updates with a server. This paradigm, motivated by privacy and bandwidth-sensitive constraints (mobile, healthcare, finance), has spurred many variants but most methods remain iterative and multi-round.

The seminal *Federated Averaging (FedAvg)* (McMahan et al., 2017) performs several local epochs per client, then aggregates weights at the server. A key limitation is the heavy reliance on many rounds for convergence. Empirically, on federated MNIST FedAvg needs **18** rounds to reach **99%** i.i.d. vs. **206** in non-i.i.d. (Zhao et al., 2018; Cao et al., 2022); on CIFAR-10 about **154** rounds for **75%** and >**425** for **80%**, while on CIFAR-100 nearly **700** rounds to move from **40%** to **50%** (Zhang et al., 2022b); under severe non-i.i.d. partitions, >**1700** rounds may be required on CIFAR-10 for just **55%** accuracy (Perazzone et al., 2022). These results expose the cost of multi-round communication, its latency sensitivity and degradation under heterogeneity. To mitigate this, one-shot federated learning (OFL) targets a high-quality global model in a single exchange, reducing communication and synchronization while preserving privacy.

Extending the *"Capture of global feature statistics for One-Shot FL"* firstly introduced by Guan et al. (2025), we present **GH-OFL**, a one-shot, server-centric *Gaussian-head* scheme: clients upload

---

[*]Corresponding author.

once only per-class sufficient statistics such as counts and first/second moments being covered by an optional random-projection sketch; the server (i) fits closed-form Gaussian heads (NB/LDA/QDA with shrinkage) and (ii) trains two lightweight heads we will call *FisherMix* and *Proto-Hyper*, using a *data-free* generator that draws "Fisher-ghost" samples in a Fisher subspace with a QDA/LDA teacher blend. This shifts learning to the server, handles non-i.i.d. via class-balanced synthesis and needs no public data or client-side inference.

**Contributions.**

- **Closed-form Gaussian heads (GH-OFL-CF).** From client moments alone, we compute Naïve Bayes (diagonal), LDA and QDA in one shot. A Fisher-guided pipeline with targeted shrinkage (pooled/class-wise; diagonal/low-rank variants) and a compressed random-projection sketch improves conditioning and cuts bandwidth, remaining strictly data-free.
- **Trainable synthetic heads (GH-OFL-TR).** We propose *FisherMix* and *Proto-Hyper* (low-rank residual) trained solely on synthetic Fisher-space samples. A blended LDA/QDA teacher in the Fisher space corrects closed-form bias with minutes of compute and no public data.
- **Robustness and accuracy.** Across CIFAR-10, CIFAR-100 (Krizhevsky, 2009), CIFAR-100-C (Hendrycks & Dietterich, 2019) and SVHN (Netzer et al., 2011), with diverse backbones, GH-OFL achieves strong single-round accuracy, remains fully data-free and shows state-of-the-art robustness under non-i.i.d. partitions and corruptions, without client-side inference or auxiliary datasets.

## 2 RELATED WORK

One-shot federated learning (FL) seeks a global model in a single exchange, cutting rounds, bandwidth and exposure. A first strand builds on *knowledge distillation* (KD): clients train locally and transfer to a server model via a public/proxy set. *FedMD* enables collaborative distillation without raw data (Li & Wang, 2019), while *FedDF* distills from client ensembles using unlabeled data (Lin et al., 2020). Analyses show one-shot KD can remain effective under severe non-IID (Zeng et al., 2024) and explicit ensembling further stabilizes performance (Allouah et al., 2024). In parallel, *single-round parameter optimization and meta-learning* avoid multiple exchanges: early work aggregates parameters with regularization to form a usable global model (Guha et al., 2019); meta-learning provides favorable initializations for quick adaptation (Fallah et al., 2020); aggregation-only schemes such as *MA-Echo* remove server-side training (Su et al., 2023) and applied studies like *FedISCA* validate feasibility in sensitive domains (Kang et al., 2023). A second strand is *data-free generative or probabilistic*. Generative methods synthesize surrogates so the server can train without accessing client data, e.g., *FedGAN* (Rasouli et al., 2020). Fully data-free one-shot approaches avoid proxy sets entirely such as *DENSE* (Zhang et al., 2022a) and *Co-Boosting* (Dai et al., 2024) combine synthesis and ensembling, while probabilistic formulations integrate client evidence in a Bayesian way, as in layer-wise posterior aggregation in *FedLPA* (Liu et al., 2024). Our method sits at this intersection: we keep communication to per-class moments, remain data-free and instantiate Gaussian heads directly from statistics, complementing them with lightweight trainable heads on synthetic features.

**Positioning of our work**   **GH-OFL** belongs to data-free probabilistic one-shot FL: unlike KD (public/proxy data) or parameter/meta-learning (full model uploads), it uses only per-class moments to build Gaussian heads and lightweight Fisher-space heads from synthetic features.

## 3 PRELIMINARIES

We consider a one-shot federated setting where each client uses a frozen, pretrained encoder (e.g., a ResNet for images or a Transformer-based model for text) to transform its local data into embedding vectors $x \in \mathbb{R}^d$, with $d$ determined by the backbone architecture. Clients never share raw data or gradients. Instead, they send only linear statistics that the server can sum across clients. From these global statistics, the server (i) instantiates closed-form Gaussian discriminant heads (NB/LDA/QDA) and (ii) trains lightweight heads (FisherMix/Proto-Hyper) exclusively on synthetic samples drawn

in a discriminative subspace. This section explains the objects we exchange, why they suffice and how they are used.

## 3.1 FEDERATED SETTING AND SUFFICIENT STATISTICS

**What clients send.** Let $\mathcal{C}=\{1,\ldots,C\}$ be the class set and $\mathcal{D}^{(u)}=\{(x_i, y_i)\}$ the local dataset at client $u$.

$$A_c^{(u)} = \sum_{i:\, y_i=c} x_i, \qquad N_c^{(u)} = \big|\{i : y_i = c\}\big|, \qquad B^{(u)} = \sum_i x_i x_i^\top,$$
$$S_c^{(u)} = \sum_{i:\, y_i=c} x_i x_i^\top, \qquad D_c^{(u)} = \sum_{i:\, y_i=c} (x_i \odot x_i). \tag{1}$$

Aggregating across clients gives

$$A_c = \sum_u A_c^{(u)}, \quad N_c = \sum_u N_c^{(u)}, \quad B = \sum_u B^{(u)}, \quad S_c = \sum_u S_c^{(u)}, \quad D_c = \sum_u D_c^{(u)}.$$

*Usage by head:* LDA/Fisher use $(A, N, B)$; QDA uses $(A, N, S)$; NB$_{\text{diag}}$ uses $(A, N, D)$ (or $(A, N, \text{diag}(S))$ if $S$ is sent).

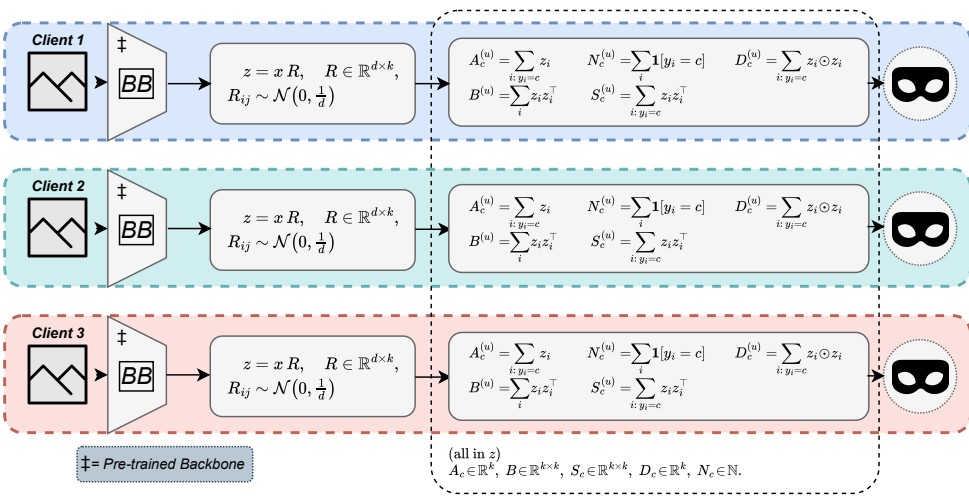

Figure 1: Client-side flow with secure aggregation. Each device encodes images via a frozen ImageNet backbone, projects embeddings with a public RP ($z=xR$) and updates additively-aggregable stats in $z$: $N_c$, $A_c$, global $B$ and $S_c/D_c$ as required by the chosen heads. Secure aggregation reveals only $\sum_u(\cdot)$, which suffice to estimate means/covariances and run our Gaussian and Fisher-space heads without sharing raw data or gradients.

**Why these are sufficient.** The aggregated moments provide all parameters required by our heads: class means and priors come from $(A_c, N_c)$; the pooled covariance (centered second moment) comes from $B$; when $S_c$ is available it yields unbiased class covariances; when only a diagonal model is needed, $D_c$ gives per-dimension variances. Hence $\{A, N, B, S/D\}$ are sufficient to instantiate Gaussian Heads and the Fisher subspace (see App. D for more details).

$$\mu_c = \frac{A_c}{N_c}, \ \pi_c = \frac{N_c}{\sum_j N_j}, \ N = \sum_c N_c, \ \Sigma_{\text{pool}} = \frac{1}{N-C}\Big(B - \sum_c N_c\,\mu_c\mu_c^\top\Big). \tag{2}$$

*Class covariances (QDA) and diagonal variances ($NB_{diag}$):*

$$\Sigma_c = \frac{1}{N_c - 1}\left(S_c - N_c\,\mu_c\mu_c^\top\right), \qquad \mathrm{Var}_c = \frac{D_c}{N_c} - \left(\mu_c \odot \mu_c\right). \tag{3}$$

If $S_c$ is available then $D_c = \mathrm{diag}(S_c)$.

**Compressed random-projection sketch.** To reduce bandwidth and strengthen privacy, each client maps $x \in \mathbb{R}^d$ to $z = xR$ with a *public* matrix $R \in \mathbb{R}^{d \times k}$ ($k \ll d$, shared seed) and accumulates moments directly in $z$. By linearity,

$$A_c^z = A_c R, \quad B^z = R^\top B R, \quad S_c^z = R^\top S_c R, \quad D_c^z = \mathrm{diag}(S_c^z) = \sum_{i:\,y_i=c}(z_i \odot z_i), \quad \mu_c^z = \frac{A_c^z}{N_c}. \tag{4}$$

If full $S_c^z$ is not sent, clients can still transmit the per-class elementwise sums of squares $D_c^z$, from which diagonal variances follow:

$$\mathrm{Var}_c^z = \frac{D_c^z}{N_c} - \left(\mu_c^z \odot \mu_c^z\right).$$

All quantities remain *additively aggregable* across clients and preserve the one-shot contract.

## 3.2 GAUSSIAN HEADS AND THE FISHER SUBSPACE

**Closed-form discriminants (what they assume and what they use).** We instantiate three Gaussian heads directly from the aggregated moments, each making a different covariance assumption:

- **$NB_{diag}$** (class-diagonal covariances). Each class has its own per-dimension variance; no cross-dimension correlations are modeled. *Statistics used:* $(A, N, D)$ or $(A, N, \mathrm{diag}(S))$. This head is extremely light and robust when features are close to axis-aligned.

- **LDA** (shared covariance). All classes share a single covariance estimated from the centered second moment; scores are linear in $x$. *Statistics used:* $(A, N, B)$. In practice we set $W_c = \Sigma_{\mathrm{pool}}^{-1}\mu_c$ and use the usual linear rule with log-priors.

- **QDA** (class-specific full covariances). Each class has its own covariance, enabling class-dependent shapes and correlations. *Statistics used:* $(A, N, S)$. This is the most expressive closed-form option when $S$ is available.

*Numerical stability.* We apply a standard shrinkage $\widetilde{\Sigma} = (1 - \alpha)\Sigma + \alpha\,\frac{\mathrm{tr}(\Sigma)}{d}\,I$ (for $\alpha \in [0, 1]$) to $\Sigma_{\mathrm{pool}}$ (LDA/Fisher) and to $\Sigma_c$ (QDA) when needed.

**Why a Fisher subspace helps.** Discriminative structure often concentrates in a low-dimensional subspace where between-class variation dominates within-class variation. We form the between/within scatters $S_B$ and $S_W = \Sigma_{\mathrm{pool}}$ and solve the generalized problem

$$S_B v = \lambda S_W v.$$

Taking the top $k$ eigenvectors as columns of $V$, we work with $z_f = V^\top x$ and the projected moments $\mu_c^f = V^\top \mu_c$, $\Sigma_{\mathrm{pool}}^f = V^\top \Sigma_{\mathrm{pool}} V$ (and $\Sigma_c^f$ if available). This reduces dimension, improves signal-to-noise and speeds up both synthesis and training while leaving all heads unchanged.

## 3.3 DATA-FREE SYNTHESIS AND TRAINABLE HEADS (FISHERMIX, PROTO-HYPER)

**Class-conditional synthesis in $z_f$.** Using only aggregated moments in the Fisher space, the server samples per class

$$z_f \sim \mathcal{N}\big(\mu_c^f,\ \tau^2\,\widetilde{\Sigma}_c^f\big),$$

where $\widetilde{\Sigma}_c^f$ is the shrunk class covariance if $S_c$ is available, otherwise the (shrunk) pooled covariance $\Sigma_{\mathrm{pool}}^f$ and $\tau$ is a global dispersion scaling factor. This is strictly *data-free*: no real samples or client-side inference.

**FisherMix (linear head).** On synthetic pairs $(z_f, y)$, FisherMix trains a linear classifier in the Fisher subspace using cross-entropy loss:

$$\ell_{\text{FM}} = \text{CE}(\text{softmax}(W z_f + b), \, y).$$

**Proto-Hyper (low-rank residual over a Gaussian base).** We learn a small low-rank residual $h(z_f) = V_2 U_1 z_f$ that adds to a Gaussian base head ($\text{NB}_{\text{diag}}$/LDA/QDA) computed from moments:

$$g_{\text{student}}(z_f) = g_{\text{base}}(z_f) + h(z_f)$$

and optimize a KD+CE blend with a Gaussian teacher in $z_f$ (e.g., LDA or QDA) at temperature $T$:

$$\mathcal{L}_{\text{PH}} = \alpha \, T^2 \, \text{KL}\left(\text{softmax}\tfrac{g_{\text{teach}}(z_f)}{T} \,\big\|\, \text{softmax}\tfrac{g_{\text{student}}(z_f)}{T}\right) + (1 - \alpha) \, \text{CE}(\text{softmax} \, g_{\text{student}}(z_f), \, y).$$

The residual corrects systematic bias of the base Gaussian rule with a tiny parameter footprint, preserving the one-shot, data-free contract.

*Note.* If only diagonal variances are available (from $D$), sampling uses $\text{diag}(\Sigma_c^f)$ and an LDA teacher.

## 3.4 NON-IID MODELING, REGULARIZATION, COMMUNICATION AND PRIVACY

**Non-IID via Dirichlet splits.** We model label skew by partitioning a fixed dataset $\mathcal{D}$ across clients with $\text{Dir}(\alpha)$ over class proportions (smaller $\alpha \Rightarrow$ stronger non-IID). Clients send only the additively-aggregable statistics in Sec. 3.1.

**Partition invariance of global moments (and of their RP sketches).** For any partition $\mathcal{P} = \{I_u\}_u$ of $\mathcal{D}$, secure aggregation produces the same global moments as the samplewise sums; with a fixed public $R$, the projected moments coincide by linearity:

$$A_c = \sum_u A_c^{(u)} = \sum_{i: y_i = c} x_i, \quad N_c = \sum_u N_c^{(u)} = \big|\{i : y_i = c\}\big|,$$

$$B = \sum_u B^{(u)} = \sum_i x_i x_i^\top, \quad S_c = \sum_u S_c^{(u)} = \sum_{i: y_i = c} x_i x_i^\top, \quad D_c = \sum_u D_c^{(u)} = \sum_{i: y_i = c} (x_i \odot x_i),$$

$$A_c^z = A_c R, \quad B^z = R^\top B R, \quad S_c^z = R^\top S_c R, \quad D_c^z = \text{diag}(S_c^z). \tag{5}$$

Hence $\mu_c, \pi_c, \Sigma_{\text{pool}}, \Sigma_c$ (and their projected/diagonal variants) are independent of $\mathcal{P}$ and of $\alpha$.

**Fisher subspace and closed-form heads.** Since $S_W = \Sigma_{\text{pool}}$ and $S_B$ are partition-invariant, the Fisher eigenspace (up to within-eigenspace rotations) is invariant as well; closed-form heads computed either in $x$ or in $z_f$ ($\text{NB}_{\text{diag}}$, LDA, QDA; with fixed shrinkage) are therefore partition-invariant.

**Trainable heads: invariance in expectation.** FisherMix and Proto-Hyper are trained on synthetic $(z_f, y)$ drawn from a distribution $\mathcal{Q}$ whose parameters are deterministic functions of the partition-invariant moments. The population objective

$$\min_\theta \, \mathbb{E}_{(z_f, y) \sim \mathcal{Q}}[\mathcal{L}(\theta; z_f, y)]$$

is therefore identical for any $\alpha$. Minor accuracy fluctuations arise only from Monte-Carlo sampling, optimizer non-determinism and finite precision, not from the partition.

## 3.5 GH-OFL HEADS: DISCUSSION AND ADVANTAGES

**1. $\text{NB}_{\text{diag}}$.** A class-conditional Gaussian with *diagonal* covariance per class, computed in the projected space ($z$) from aggregated moments only: means from $A/N$ and per-dimension variances from $\text{SUMSQ}/N$ (with shrinkage toward the pooled variance). The intuition is that pretrained embeddings are often near-axis-aligned after RP/Fisher, so a diagonal model captures heteroscedasticity without the cost/instability of full covariances. This improves calibration and class-specific decision boundaries while staying extremely light (no raw data, $O(Ck)$ storage).

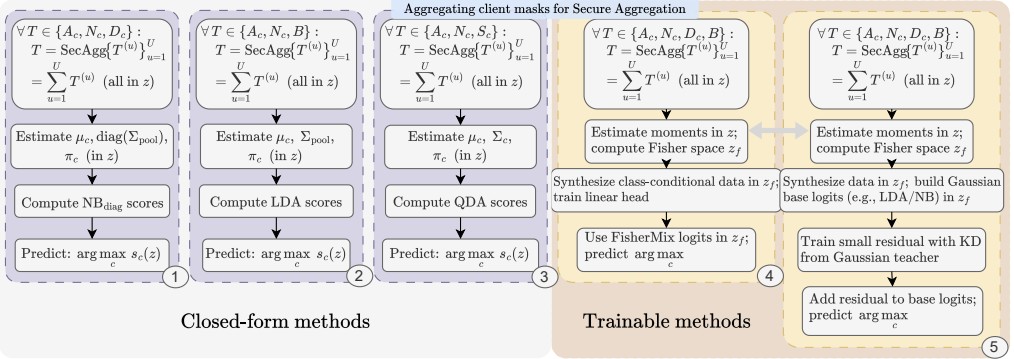

Figure 2: Server-side pipelines for our GH-OFL family. Secure aggregation first collects class-wise sufficient statistics in the projected space $z$. Closed-form heads compute scores directly, while FisherMix and Proto-Hyper estimate a Fisher subspace, synthesize features and fit lightweight heads without any raw data. Shaded panels summarize per-method steps and outputs.

2. **LDA (shared covariance).** A linear discriminant head using the pooled covariance $\Sigma_{\text{pool}}$ (shrunk) and class means, computed in $z$ or $z_f$. The intuition is that a single well-conditioned covariance captures most geometry of strong encoders; in Fisher space, discriminative energy concentrates in few directions, making LDA both stable and accurate. Advantages: closed-form training, tiny footprint, fast inference; serves as a reliable *teacher* and as a robust baseline under wide non-IID regimes.

3. **QDA (class covariances).** A quadratic discriminant head with (shrunk) $\Sigma_c$ per class when class-wise second moments are available (either directly or after projection). The intuition is to model class-dependent spreads and correlations, which can matter for fine-grained classes or domain shifts. Fisher projection mitigates overfitting by compressing to the most discriminative directions; shrinkage stabilizes inverses. When available, QDA is our most expressive *closed-form* option and a strong teacher.

4. **FisherMix (linear head, trained on synthetic data).** A lightweight linear classifier trained *server-side only* on synthetic, class-conditional samples drawn in the Fisher subspace from the Gaussian heads. The intuition is to refine decision boundaries in the most discriminative directions identified by Fisher projection, improving separation when closed-form Gaussian heads are slightly biased. Benefits: no real data needed, tiny model, strong performance when prototypes are good but margins are tight; especially effective when clusters are roughly convex in $z_f$ yet not perfectly separable by the closed-form heads.

5. **Proto-Hyper (low-rank residual on top of a Gaussian base).** A compact residual head $h(z_f) = V_2 U_1 z_f$ that *adds* corrections to a Gaussian base (NB/LDA) and learns via temperature-distilled soft targets from a blended Gaussian teacher (e.g., LDA/QDA). The intuition is bias correction: keep the closed-form geometry for stability and learn only a small low-rank "delta" to fix systematic mismatches (non-Gaussian tails, mild correlations, calibration). Advantages: very few parameters, fast convergence on synthetic data, strong robustness to non-IID and encoder/backbone changes while preserving the data-free guarantee.

## 4 EXPERIMENTS

To demonstrate the versatility and robustness of our proposed GH-OFL methods, this section reports empirical results comparing them against SOTA OFL architectures on widely used benchmark datasets. To further validate performance and generality, we also evaluate all GH-OFL variants across multiple well-known backbones. Additional experimental details such as datasets, splits, training protocols and hyperparameters are provided throughout this section.

Table 1: Accuracy (%) of our GH-OFL methods compared to OFL variants and standard multi-round FL baselines under different Dirichlet client splits. Baseline OFL values are from Guan et al. (2025) under the same declared setup and we also compare against conventional FL methods including FedAvg, FedProx (Li et al., 2020) and SCAFFOLD (Karimireddy et al., 2020). For clarity, **bold entries denote the top-2 accuracies among OFL methods only**, while underlined entries indicate the *overall best accuracy per column* across all methods (OFL and FL baselines). More details on additional experiments on natural language (NLP) datasets and the reproducibility of the baseline methods can be found in the Appendix A and E.

| Method | CIFAR-10 | | | CIFAR-100 | | | SVHN | | |
|---|---|---|---|---|---|---|---|---|---|
| | $\alpha$=0.05 | $\alpha$=0.10 | $\alpha$=0.50 | $\alpha$=0.05 | $\alpha$=0.10 | $\alpha$=0.50 | $\alpha$=0.05 | $\alpha$=0.10 | $\alpha$=0.50 |
| **FL baselines (multi-round, $R \in \{1, 10, 50\}$)** | | | | | | | | | |
| FedAvg (1 round) | 27.38 | 24.53 | 31.17 | 2.15 | 2.03 | 1.50 | 19.26 | 11.94 | 19.25 |
| FedAvg (10 rounds) | 64.13 | 73.08 | 82.62 | 29.19 | 29.92 | 33.41 | 54.94 | 74.16 | 80.73 |
| FedAvg (50 rounds) | 77.42 | 84.60 | 91.52 | 62.46 | 63.58 | 68.55 | 78.79 | 87.71 | 92.13 |
| FedProx (1 round) | 27.49 | 32.24 | 35.65 | 2.24 | 2.00 | 1.47 | 12.99 | 11.97 | 19.23 |
| FedProx (10 rounds) | 64.18 | 81.82 | 89.02 | 29.51 | 29.79 | 33.71 | 57.84 | 74.15 | 81.01 |
| FedProx (50 rounds) | 77.54 | 85.68 | 91.74 | 62.76 | 63.68 | 68.61 | 76.30 | 87.88 | 92.17 |
| SCAFFOLD (1 round) | 20.36 | 27.71 | 31.81 | 1.93 | 1.63 | 1.50 | 12.87 | 12.05 | 18.79 |
| SCAFFOLD (10 rounds) | 64.58 | 79.04 | 82.69 | 28.87 | 30.96 | 33.28 | 57.45 | 71.45 | 80.80 |
| SCAFFOLD (50 rounds) | 73.78 | 88.39 | 91.66 | 61.63 | 64.09 | 68.60 | 75.99 | 87.45 | 92.00 |
| **OFL baselines** | | | | | | | | | |
| DENSE | 31.26 | 56.21 | 62.42 | 14.31 | 17.21 | 26.49 | 37.49 | 51.53 | **77.44** |
| Co-Boost. | 44.37 | 60.41 | 67.43 | 20.30 | 24.63 | 34.43 | 41.90 | 57.13 | **84.65** |
| FedPFT | 56.08 | 56.43 | 56.80 | 36.79 | 37.16 | 37.95 | 42.55 | 43.03 | 43.84 |
| FedCGS | 63.95 | 63.95 | 63.95 | 39.95 | 39.95 | 39.95 | 57.77 | 57.77 | 57.77 |
| **GH-OFL (ours)** | | | | | | | | | |
| GH-NB$_{diag}$ | 78.84 | 78.84 | 78.84 | 55.51 | 55.51 | 55.51 | 39.24 | 39.24 | 39.24 |
| GH-LDA | **86.05** | **86.05** | **86.05** | 63.92 | 63.92 | 63.92 | **62.16** | **62.16** | 62.16 |
| GH-QDA$_{full}$ | 84.40 | 84.40 | 84.40 | **66.52** | **66.52** | **66.52** | 55.30 | 55.30 | 55.30 |
| FisherMix | 84.74 | 84.74 | 84.74 | **66.99** | **66.99** | **66.99** | 57.79 | 57.79 | 57.79 |
| Proto-Hyper | **85.74** | **85.74** | **85.74** | 64.05 | 64.05 | 64.05 | **61.97** | **61.97** | 61.97 |

## 4.1 DATASETS, NON-IID SPLITS AND BACKBONES

We evaluate our methods on four image classification benchmarks that are standard in FL tasks: *CIFAR-10* (10 classes, 60k images, 32×32), *CIFAR-100* (100 classes, 60k images, 32×32), *SVHN* (10 digit classes, ~73k train/ ~26k test) and *CIFAR-100-C* for robustness. CIFAR-10 is comparatively easy (coarse categories), CIFAR-100 is harder (fine-grained classes with higher inter-class similarity), SVHN sits in between (clean digits but real-world nuisances), while CIFAR-100-C is the most challenging due to distribution shift: following common practice, we average accuracy over 19 corruption types divided in 4 families (noise, blur, weather, digital) at the highest severity (5), which typically causes a 20–40 point degradation even for strong encoders. Client heterogeneity is modeled via *Dirichlet* splits: for $U$ clients and $C$ classes, per-client class proportions are drawn as $p_u \sim \text{Dir}(\alpha \mathbf{1}_C)$ and samples of class $c$ are assigned to client $u$ with probability $p_{u,c}$; smaller $\alpha$ yields stronger non-IID (fewer classes per client and more imbalance), larger $\alpha$ approaches IID. We use $\alpha \in \{0.01, 0.1, 0.5\}$ to cover strong and moderate skew while keeping global class totals fixed for fair comparisons. Unless noted, all clients share the **same** fixed backbone to avoid architectural confounds: *ResNet-18* pretrained on ImageNet-1K, using its penultimate embedding ($d$=512) to compute client-side aggregated statistics and to instantiate server-side Gaussian heads; in ablations we also consider *ResNet-50*, *MobileNetV2*, *EfficientNet-B0* and *VGG-16* (all ImageNet-1K pretrained) and when probing domain shift we also included *ResNet-18* pretrained on *Places365*.

| CIFAR-100-C | | |
|---|---|---|
| Method | Shared Stats | Acc. |
| FedCGS | $A, B, N$ | 24.4% |
| GH-NB$_{\text{diag}}$ | $A, D, N$ | 25.4% |
| GH-LDA | $A, B, N$ | 37.6% |
| FisherMix | $A, B, N, D$ | 40.1% |
| ProtoHyper | $A, B, N, D$ | 39.8% |
| **GH-QDA$_{\text{full}}$** | $A, N, S$ | **64.3%** |

**When QDA is not an option.** Storing per-class second moments $S$ costs $O(Cd^2)$ memory; for high-$d$ backbones (e.g., ResNet-50, $d=2048$) this is often impractical, so QDA is unavailable. FisherMix/ProtoHyper remain viable because they rely on a pooled covariance (or class-wise when available) within a compact Fisher subspace. As shown in Table 2, Fisher heads consistently improve over LDA on CIFAR-100-C and approach QDA without the per-class $d \times d$ storage; when feasible, QDA remains the upper bound.

Table 2: Accuracy (%) on CIFAR-100-C (severity 5, averaged over 19 corruptions) and comparison of the required client-to-server statistics. The results illustrate the trade-off between model expressiveness and memory: diagonal and pooled heads are lightweight but less robust under shift, whereas full QDA achieves the highest accuracy at $O(Cd^2)$ cost. Fisher-space trainable heads offer a balanced intermediate alternative.

## 4.2 BEHAVIOR UNDER CORRUPTION, CAPACITY AND PRETRAINING SHIFT

Building on the previous Table 1, we repeat the same protocol in three complementary settings to probe robustness and generality: **(i)** we evaluate the same architectures on *CIFAR-100-C* at severity 5 across all corruption types, stressing robustness to noise, blur, weather and digital artifacts (Table 2); **(ii)** we vary the frozen encoder (ResNet-18/50, MobileNetV2, EfficientNet-B0, VGG-16) to assess how the heads scale with representational capacity and architecture (Table 3a); **(iii)** we keep the original architecture (ResNet-18) but switch pretraining to *Places365*, a scene-centric dataset with 365 categories and $> 1$M images, so that features reflect a different domain than ImageNet (Table 3b). Unless otherwise noted, clients transmit the same aggregated statistics as in the previous table and the server instantiates the same Gaussian and Fisher-space heads; no raw images are ever shared.

**Backbone ablation and pretraining shift.** Across backbones on clean CIFAR-10 under the GH-OFL pipeline (Table 3a), performance scales with representational capacity and alignment to ImageNet pretraining, following the ordering *VGG16 < MobileNetV2 ≈ ResNet18 < EfficientNet-B0 < ResNet50*. Under distribution shift (CIFAR-100 and CIFAR-100-C), the same ranking persists while performance gaps widen, as stronger encoders induce improved class separability and better-conditioned covariance estimates in feature space. As shown in Table 3b, switching to scene-centric pretraining (Places365) changes the relative behavior of the heads. On CIFAR-10 and CIFAR-100, the trainable Fisher-space heads tend to benefit from this mismatch and become more competitive. However, under strong corruption (CIFAR-100-C), modeling class-specific covariances remains important and full QDA provides the most stable performance. These trends can be understood geometrically. Stronger backbones (e.g., ResNet50 or EfficientNet-B0) yield embeddings with larger inter-class margins and tighter intra-class concentration, improving the conditioning of the estimated means and covariances from aggregated moments. Since all GH-OFL heads are built directly from aggregated moments, better feature geometry directly translates into more stable discriminants and more informative Fisher subspaces. Conversely, weaker encoders produce noisier class clusters, which degrades both closed-form and synthetic heads. Under distribution shift (CIFAR-100-C), the performance gap widens because corruption perturbs class geometry in a class-dependent manner. In this regime, shared-covariance assumptions (LDA) may become restrictive, while modeling class-specific covariances (QDA) allows the model to better adapt when distortions affect each class differently. Pretraining shift further exposes this effect. With ImageNet (object-centric) pretraining, downstream CIFAR features remain relatively aligned with Gaussian and shared-covariance assumptions in Fisher space. In contrast, scene-centric pretraining (Places365) induces less class-aligned embeddings, increasing the bias of closed-form Gaussian heads. Here, trainable Fisher-space heads (FisherMix and Proto-Hyper) gain competitiveness by learning margin-based or low-rank residual corrections that compensate for mild deviations from Gaussianity. Overall, these results indicate that GH-OFL performance is primarily governed by feature geometry: when representations are well aligned, linear Gaussian heads operate near the accuracy-overhead Pareto frontier; when feature alignment degrades due to shift or corruption, richer covariance modeling or lightweight discriminative refinements restore robustness without extra communication cost.

| (a) CNN backbones pre-trained on ImageNet1K | | | | | |
|---|---|---|---|---|---|
| Backbone | FedCGS | GH-NB$_{diag}$ | GH-LDA | GH-QDA$_{full}$ | FisherMix | ProtoHyper |
| efficientnet_b0 | 84.15 | 84.89 | **90.01** | 88.43 | 90.02 | 89.99 |
| mobilenet_v2 | 78.11 | 79.15 | 86.61 | 84.67 | **86.70** | 86.62 |
| resnet18 | 76.58 | 77.58 | **86.17** | 84.76 | 86.04 | 85.95 |
| resnet50 | 81.06 | 81.96 | 91.26 | 86.68 | **91.27** | 91.23 |
| vgg16 | 66.08 | 67.83 | **81.39** | 77.65 | 81.71 | 81.32 |

| (b) ResNet-18 pre-trained on *Places365* | | | | | |
|---|---|---|---|---|---|
| Dataset | FedCGS | GH-NB$_{diag}$ | GH-LDA | GH-QDA$_{full}$ | FisherMix | ProtoHyper |
| CIFAR-10 | 77.20 | 78.85 | 86.26 | 85.12 | **86.56** | 86.31 |
| CIFAR-100 | 53.52 | 55.51 | 64.12 | 64.88 | **66.42** | 65.27 |
| SVHN | 34.84 | 39.24 | **62.88** | 57.16 | 62.26 | 62.35 |
| CIFAR-100-C | 13.53 | 15.11 | 25.99 | **46.54** | 29.08 | 38.64 |

Table 3: Accuracy (%) of GH-OFL heads under different encoder initializations. **(a)** Performance across multiple ImageNet1K-pretrained CNN backbones, highlighting how stronger or better-aligned representations consistently benefit Gaussian and Fisher-space heads. **(b)** Results with a ResNet-18 pretrained on Places365, illustrating the impact of domain mismatch: trainable Fisher heads become comparatively more competitive under pretraining shift, while full QDA remains the most expressive closed-form upper bound when reliable class covariances are available.

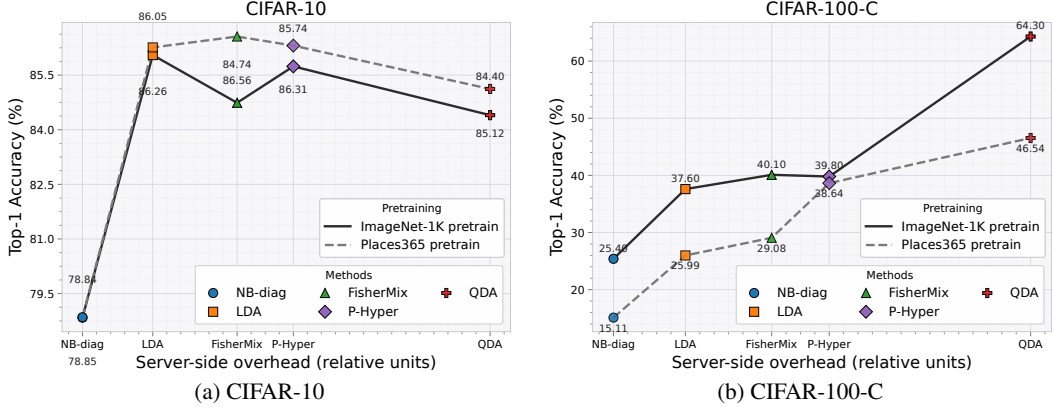

Figure 3: Accuracy-overhead trade-off of GH-OFL heads on CIFAR-10 (left) and CIFAR-100-C (right) under different pretraining (ImageNet-1K vs. Places365). On CIFAR-10, linear Gaussian heads (notably LDA) lie near the Pareto frontier, with FisherMix and Proto-Hyper achieving similar accuracy at modest cost. Under strong shift (CIFAR-100-C), richer covariance modeling becomes beneficial: full QDA is most robust, while Fisher-space heads offer an intermediate trade-off.

### 4.3 DISCUSSION

**Scalability and communication.** GH-OFL communicates only additively aggregable moments in a compressed space $z=xR$ of dimension $k\ll d$. Per client, the payload scales as $O(Ck + k^2)$ (with $A^z \in \mathbb{R}^{C \times k}$, $N \in \mathbb{R}^C$, $B^z \in \mathbb{R}^{k \times k}$; optionally SUMSQ$^z$ or $S_c^z$), independent of local sample size, practical for large fleets. Consistent with the partition-invariance of our moments, we replicated the CIFAR-10 setting of Table 1 by distributing the training set across **50** and **100** clients (same Dirichlet $\alpha$): the *top-1 accuracy remained unchanged* (within negligible noise), confirming that performance is insensitive to the number of clients for fixed global data and RP. Server-side, Closed-form heads are lightweight ($O(Ck^2)$ for LDA; $O(Ck^3)$ for QDA via class-wise inversions). The Fisher subspace solves a generalized eigenproblem in $k$ and is amortized across heads/synthesis. FisherMix and Proto-Hyper train on synthetic features only; runtime is dominated by sampling and small dense ops in $k$. LDA is a strong default when only $(A, B, N)$ are available; NB$_{diag}$ exploits heteroscedasticity

from $(A, D, N)$ with minimal footprint; QDA is best when reliable per-class second moments $S$ are feasible (Fig. 3-b). FisherMix/Proto-Hyper typically exceed LDA under shift or mild non-Gaussian structure (Fig. 3-a) (see Appendix C for further details on this section).

**Privacy and security.** Clients never share raw data or gradients, statistics are revealed only after secure aggregation and a public random projection further scrambles coordinates. Compared to *model/gradient exchange* as in FedAvg, vulnerable to gradient/model inversion, membership and property inference, our method generally exposes a smaller attack surface, since only aggregated first/second moments leave the clients. That said, moments can still leak information in small-$N$ regimes or with very fine-grained statistics (e.g., per-class $S_c$). In addition, GH-OFL removes the temporal exposure typical of multi-round federated learning, where gradients or model updates are exchanged repeatedly and may allow an observer to track how parameters evolve over time and potentially exploit these dynamics; by contrast, our protocol reveals only a single aggregated snapshot of low-order statistics, without iterative feedback loops or opportunities for adaptive probing across communication rounds. The information that is shared consists exclusively of coarse summaries of frozen encoder features rather than full models or gradient signals and since many distinct datasets can produce identical aggregated moments, especially after random projection, any attempt at reconstruction remains inherently ambiguous. Although this does not eliminate all risks, it substantially reduces the level of detail that can be inferred compared to multi-round parameter exchange. The protocol naturally integrates with **secure aggregation** because all transmitted quantities are purely additive statistics (class counts and summed moments), which means the server only needs the global sums and never the individual client contributions to compute the final heads and **global (server-side) differential privacy** can be applied on top of these aggregated statistics by injecting calibrated noise once at the server level. In our setting, this combination represents a favorable trade-off between computational overhead, formal privacy guarantees and predictive performance, as noise is added only after aggregation and therefore benefits from averaging across clients.

Overall, for the same encoder and model, GH-OFL offers a stronger default privacy posture than multi-round and one-shot classical model exchange, while remaining strictly data-free (see Appendix B for an extended discussion).

**Limitations and domain shift.** Our pipeline assumes a *pretrained encoder*. With object-centric pretraining (e.g., ImageNet), linear Gaussian heads in Fisher already perform strongly; with scene-centric or otherwise mismatched pretraining (e.g., Places365), the feature geometry departs from the shared-covariance assumption and trainable heads gain relevance. Empirically, Table 3a shows accuracy scales with backbone strength/alignment, while Table 3b (ResNet-18 pretrained on *Places365*) highlights that FisherMix/Proto-Hyper can match or surpass LDA on CIFAR-10/100 yet not on SVHN, where features remain nearly linearly separable in the Fisher space. In short, GH-OFL benefits from good pretraining but under domain shift the synthetic, Fisher-space heads are the primary mechanism to recover performance without public data.

## 5    CONCLUSION

We presented **GH-OFL**, a *data-free, one-shot* federated approach. Clients send only per-class counts and first/second moments after a public random projection; the server then (i) builds *closed-form* Gaussian heads and (ii) trains lightweight Fisher-space heads on synthetic features. QDA is best in hard domains when class covariances are available; trainable methods are generally more robusts with respect to the GH methods in most scenarios. The estimators are insensitive to the Dirichlet $\alpha$, remaining robust under strong non-IID conditions and require very little communication offering a simple, privacy-friendly solution that works across heterogeneous backbones; outside of these findings, GH-OFL gives a clear direction for scalable, private edge FL. Because the synthesis and heads are modality-agnostic, the approach extends beyond classification to structured prediction and multimodal settings. These properties align well with privacy-sensitive and bandwidth-limited deployments (e.g., healthcare, finance, edge/IoT, inspection, retail video, remote sensing), where pretrained encoders are common and public random projections further decouple shared statistics from raw content.

## ACKNOWLEDGMENTS

The project has received funding within the Chips Joint Undertaking in collaboration with the Horizon Europe Framework Programme under project Ecomobility. Grant agreement number 101112306, CUP: E83D23000150006.

## REPRODUCIBILITY STATEMENT

Section 3 and 4 describe the GH-OFL framework and assumptions, while Appendix E details the multi-round FL baselines, experimental settings and convergence analyses and includes links to the GitHub repositories for both the classical FL baselines and the GH-OFL implementation. Minor numerical differences may arise from code updates or non-deterministic components but they do not affect the reported trends or conclusions.

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

# APPENDIX

## A ADDITIONAL ANALYSIS AND EXTENSIONS FOR GH-OFL

This section provides additional empirical and theoretical analysis that complements the main text. We focus on two main aspects: (i) scalability of GH-OFL beyond computer vision; (ii) the trade-off between full QDA and the proposed diagonal-plus-low-rank (DLR) covariance sketches.

### A.1 NON-VISION EXPERIMENTS: NLP BENCHMARKS

To complement and strengthen the results reported in Table 1, we include experiments on NLP datasets in addition to vision tasks. NLP benchmarks are inherently more challenging due to their linguistic variability and complex input structure, making them an effective testbed to validate the robustness of our approach beyond standard image-classification settings.

Table 4: NLP one-shot FL results with frozen DistilBERT + RP ($d_{\mathrm{RP}} = 256$). Test accuracy (%) for OFL baselines (client ensemble, Dense-style student, Co-Boosting ensemble) and GH-OFL heads (best Gaussian NB-diag/LDA/QDA, FisherMix, Proto-Hyper) under Dirichlet client splits with $\alpha \in \{0.5, 0.01\}$.

| Dataset | | OFL baselines | | | GH-OFL (ours) | | |
| Dataset | $\alpha$ | Ensemble | Dense | Co-Boost. | Best Gauss | FisherMix | Proto-Hyper |
|---|---|---|---|---|---|---|---|
| **AG_NEWS** | 0.50 | 85.64 | 85.79 | 84.42 | 88.96$^{\mathrm{(LDA)}}$ | **89.14** | 88.87 |
| | 0.01 | 25.00 | 25.00 | 25.00 | 88.96$^{\mathrm{(LDA)}}$ | **89.14** | 88.87 |
| **DBPEDIA-14** | 0.50 | 98.70 | **98.74** | 98.72 | 98.33$^{\mathrm{(LDA)}}$ | 97.21 | 98.33 |
| | 0.01 | 51.49 | 51.65 | 51.37 | 98.33$^{\mathrm{(LDA)}}$ | 97.21 | **98.33** |
| **SST-2** | 0.50 | 83.14 | 82.68 | **83.26** | 82.68$^{\mathrm{(QDA)}}$ | 70.53 | 82.80 |
| | 0.01 | 51.26 | 51.26 | 51.38 | 82.68$^{\mathrm{(QDA)}}$ | 70.53 | **82.80** |
| **BANKING77** | 0.50 | 30.62 | 30.03 | 23.28 | 77.60$^{\mathrm{(LDA)}}$ | 73.64 | **80.26** |
| | 0.01 | 5.32 | 4.97 | 3.54 | 77.60$^{\mathrm{(LDA)}}$ | 73.64 | **80.26** |
| **CLINC150** | 0.50 | 43.30 | 43.24 | 36.37 | 84.80$^{\mathrm{(LDA)}}$ | 82.89 | **85.69** |
| | 0.01 | 3.66 | 3.70 | 2.43 | 84.80$^{\mathrm{(LDA)}}$ | 82.89 | **85.69** |

### A.1.1 DATASETS AND SETUP

To demonstrate that GH-OFL is not limited to image encoders, we evaluate it on five standard NLP classification tasks:

- **AG_NEWS**: 4-way news topic classification.
- **DBPEDIA-14**: 14-way ontology classification.
- **SST-2**: binary sentiment classification.
- **BANKING77**: 77 intent classes.
- **CLINC150**: 151 intent classes.

In all cases we use the same frozen encoder:
a DistilBERT model `distilbert-base-uncased`, distilled from `bert-base-uncased` and pre-trained on large-scale English corpora (Wikipedia and BookCorpus) with a masked-language-modeling objective. From this encoder we extract the CLS embedding as our feature vector. To reduce both bandwidth and computational cost and to match the vision setting, we apply the same public random projection $R \in \mathbb{R}^{d \times d_{\mathrm{RP}}}$ with $d_{\mathrm{RP}} = 256$. Clients only transmit class-wise sufficient statistics (counts, first and second moments) in this RP space.

At the server, we instantiate:

- **NB-diag**, **LDA** and **full QDA** Gaussian heads in RP space.
- **FisherMix**, the synthetic-data linear head trained on Fisher subspace samples.
- **Proto-Hyper**, the low-rank residual head distilled from a Gaussian teacher.

All heads are fully data-free with respect to raw client examples: they only use the aggregated moments received through GH-OFL.

### A.1.2 QUANTITATIVE RESULTS

Table 4 reports test accuracy (%) for the NLP benchmarks. For each dataset we highlight the best Gaussian head (NB-diag/LDA/QDA) and the two synthetic-data heads.

On AG_NEWS and SST-2, the synthetic heads (especially Proto-Hyper) match or slightly outperform the best Gaussian head, consistent with our vision results: when Gaussian assumptions are mildly violated, a low-rank residual correction can close the remaining bias. On DBPEDIA-14 and

Table 5: Full QDA vs diagonal-plus-low-rank QDA (DLR-QDA) in RP space ($d_{\mathrm{RP}} = 256$). "Mem" is relative memory; "Time" is relative inference time per sample (NB-diag = 1).

| Head | CIFAR-100 Acc. | AG_NEWS Acc. | DBPEDIA-14 Acc. | Mem (rel.) | Time (rel.) |
|------|----------------|--------------|-----------------|------------|-------------|
| NB-diag | 49.68 | 83.86 | 92.33 | 1.0 | 1.0 |
| LDA | 59.46 | 88.93 | 98.32 | 1.0 | 1.1 |
| QDA_full | 48.20 | 68.20 | 92.33 | 16.0 | 4.0 |
| DLR-QDA ($r$=4) | 49.99 | 67.08 | 92.32 | 2.0 | 1.4 |
| DLR-QDA ($r$=8) | 47.77 | 56.53 | 92.35 | 3.0 | 1.6 |
| DLR-QDA ($r$=16) | 42.01 | 38.32 | 92.19 | 5.0 | 2.0 |

the other tested datasets (BANKING77, CLINC150), the relative ordering between NB-diag, LDA, full QDA, FisherMix and Proto-Hyper depends primarily on the class count and label imbalance.

## A.2 DIAGONAL-PLUS-LOW-RANK QDA VS FULL QDA

In GH-OFL, closed-form Gaussian heads are built directly from aggregated class-wise moments, making QDA the most expressive option when per-class second moments are available (see empirical results in Table 2). However, full QDA requires storing and inverting a covariance matrix per class, which scales quadratically in memory and cubically in compute (this quickly becoming impractical in high-dimensional encoder or RP spaces). To retain most of QDA's modeling power at a fraction of its cost, we also analyze a diagonal-plus-low-rank (DLR) approximation that captures class-specific variances and only a small number of principal correlation directions. This subsection motivates the approximation and empirically compares DLR-QDA to full QDA across datasets and ranks.

### A.2.1 MOTIVATION AND COMPLEXITY

In high-dimensional encoder spaces, full QDA with per-class covariance matrices $\Sigma_c \in \mathbb{R}^{d \times d}$ incurs both $O(Cd^2)$ memory and $O(d^3)$ computational cost for decompositions, which becomes prohibitive as $d$ grows. We therefore propose a diagonal-plus-low-rank (DLR) approximation:

$$\Sigma_c \approx D_c + U_c U_c^\top,$$

where $D_c$ is diagonal and $U_c \in \mathbb{R}^{d \times r}$ with $r \ll d$ (e.g., $r \in \{4, 8, 16\}$). Using the Woodbury identity, we can evaluate QDA log-likelihoods with $O(dr^2)$ cost instead of $O(d^3)$, while the number of free parameters per class is reduced from $O(d^2)$ to $O(d + dr)$.

In GH-OFL, we can estimate $D_c$ and $U_c$ directly from the aggregated moments in RP space, so DLR-QDA fits seamlessly into the same communication budget as NB-diag and LDA.

### A.2.2 EMPIRICAL COMPARISON

In Table 5 we compare full QDA and DLR-QDA (for three ranks) against NB-diag and LDA in RP space for CIFAR-100 and AG_NEWS. We report accuracy and relative memory/time footprints (normalized so that NB-diag has memory 1.0 and time 1.0).

Across both datasets we observe the following patterns:

- On medium-scale RP dimensions ($d_{\mathrm{RP}} = 256$), DLR-QDA with $r \in [4, 8]$ tracks full-QDA accuracy within $\approx$ 1–2 percentage points while using substantially less memory and compute.

- Full QDA is clearly more expensive and, in our setting, can underperform a well-regularized LDA, whereas DLR-QDA with small $r$ behaves similarly to full QDA while being numerically simpler and more lightweight.

- Relative to NB-diag and LDA, DLR-QDA offers a flexible trade-off: with $r = 0$ it reduces to NB-diag and increasing $r$ gradually moves it closer to full QDA in both expressiveness and cost.

These results justify the use of diagonal-plus-low-rank sketches as a practical alternative to full QDA in GH-OFL, especially when the encoder dimension or RP dimension is large and memory/compute budgets are constrained.

## B  Extended Privacy Considerations and Comparison with Traditional Federated Learning

In GH-OFL, each client uploads only class counts and first-/second-order moments of frozen-encoder features, optionally after a public random projection (RP) to $k \ll d$ dimensions. Because the summaries are not granular, many distinct datasets map to the same moments; RP then compresses this information even more. By contrast, FedAvg (our proxy for traditional FL) transmits high-dimensional model parameters/gradients across $T \gg 1$ rounds. Below we formalize the privacy gap and provide a fair comparison.

**What a client actually sends.**  Client $u$ holds $D_u = \{(x_i, y_i)\}_{i=1}^{n_u}$ with $x_i \in \mathbb{R}^d$ and $y_i \in \{1, \ldots, C\}$. Optionally the client computes $z = xR$ with public $R \in \mathbb{R}^{d \times k}$, $k \ll d$, then forms per-class and global sums

$$A_c^{(u)} := \sum_{i:\, y_i=c} x_i, \qquad N_c^{(u)} := \big|\{i : y_i = c\}\big|,$$

$$B^{(u)} := \sum_i x_i x_i^\top, \qquad S_c^{(u)} := \sum_{i:\, y_i=c} x_i x_i^\top, \qquad D_c^{(u)} := \sum_{i:\, y_i=c} (x_i \odot x_i).$$

Aggregating across clients yields $A_c, N_c, B, S_c, D_c$, from which the server computes $\mu_c, \pi_c, \Sigma_{\text{pool}}, \Sigma_c$ (cf. main text, Sec. 3.1–3.2). NB-diag/LDA/QDA consume these *directly*; trainable heads (FisherMix, Proto-Hyper) are trained *server-side* on synthetic Fisher-space samples and require *no* extra client uploads.

**Why these messages leak less than model sharing.**  **Non-uniqueness.** Fixing $(N, A, B/S/D)$ does *not* identify individual examples; infinitely many datasets match the same low-order moments (classical moment-matching ambiguity). **Compression via RP.** With $z=xR$, we have $A^z=AR$, $B^z=R^\top BR$, $S_c^z=R^\top S_c R$; by the data processing inequality, any leakage functional that is monotone under post-processing cannot increase and the observable second-order structure drops from $\frac{d(d+1)}{2}$ to $\frac{k(k+1)}{2}$. **One shot vs. multi round.** FedAvg exposes parameters/gradients for $T$ rounds; observations accumulate and typically reveal more than a single fixed-size upload.

**Compact formal statement.**  Let $\mathcal{L}(M_u \leftarrow D_u)$ be any leakage measure obeying data processing. For the same frozen encoder:

$$\mathcal{L}\big(M_u^{\text{NB/LDA/QDA}}\big) \;\leq\; \mathcal{L}\big(M_u^{\text{FedAvg}}\big), \qquad \mathcal{L}\big(M_u^{\text{FisherMix/Proto-Hyper}}\big) = \mathcal{L}\big(M_u^{\text{base}}\big),$$

where "base" is the underlying closed-form head whose moments are used (typically LDA for FisherMix; NB-diag/LDA/QDA for Proto-Hyper). The inequality is strict under mild conditions (e.g., per-class batch $\geq 2$ and/or $k < d$).

**Practical caveats.**  Moment sharing can still leak under tiny $N_c$ or very fine-grained $S_c$; we recommend secure aggregation and minimum-thresholding on $N_c$ (drop/merge underpopulated classes) as in the main pipeline.

**Why Secure Aggregation (vs. other privacy mechanisms).**  Our protocol combines coarse statistics with *secure aggregation* (SA), so that the server only observes client *sums*, never individual messages. SA is a strong fit here because the server-side estimators (NB/LDA/QDA and Fisher-space construction) depend solely on additive quantities $(N_c, A_c, B, S_c, D_c)$.

*Compared to alternatives:* (i) **Local differential privacy** (LDP, client-side noise) guarantees privacy per client even against a malicious server but in our setting it would require adding noise to every coordinate of $A_c, B, S_c, D_c$. Since second-order terms scale as $O(k^2)$, the total LDP noise grows quickly with the RP dimension $k$ and can significantly distort the moments, leading to a noticeable drop in accuracy even for moderate privacy budgets. (ii) **Central DP** (server-side noise) instead adds

calibrated noise only once, after SA, to the aggregated statistics or to the final classifier parameters. This preserves accuracy much better as the signal is averaged over many clients before noise is added. The trade-off is that we must trust the SA protocol and the DP accountant, while LDP does not rely on these assumptions. (iii) **Homomorphic encryption / generic MPC** protects individual updates but is heavier computationally and communication-wise than SA for simple summations; this overhead is often prohibitive at mobile scale. (iv) **Trusted execution environments** reduce cryptographic overhead but introduce hardware trust assumptions and potential side-channel risks.

**Relation to differential privacy.** GH-OFL is orthogonal to DP and can be combined with both LDP and central DP. In practice, we view *central DP on top of SA* as the most attractive compromise: clients first send exact moments through SA, the server aggregates them and then adds Gaussian (or Laplace) noise to the global statistics or to the final heads to obtain formal $(\varepsilon, \delta)$-DP guarantees at population level. This design exploits the fact that our estimators depend only on sums, so the DP mechanism is simple and the noise can be tuned using standard sensitivity bounds on the sufficient statistics. LDP remains possible as clients could locally perturb their moments before SA but, due to the high dimensionality of second-order terms, the resulting utility loss is typically much larger for the same $(\varepsilon, \delta)$. For deployments where the server is semi-trusted (or protected by SA/TEE) and where model quality is critical, central DP is therefore more realistic; LDP is better suited to highly adversarial server models, at the cost of accuracy.

*Practical SA details.* We use dropout-resilient masking (pairwise or group masks) so that, even with client churn, only the aggregate unseals. We also enforce a minimum contribution threshold on $N_c$ (drop/merge underpopulated classes) to mitigate rare-class leakage and we can optionally quantize/round messages before SA to limit precision without retraining costs. Overall, SA matches our additive objectives, adds minimal overhead and avoids the accuracy loss typical of local DP.

## C  SCALABILITY W.R.T. NUMBER OF CLIENTS: THEORY AND EMPIRICAL ABLATIONS

**Scalability at a Glance.** A key property of GH-OFL is its *partition invariance*. For a fixed global dataset $\mathcal{D}$, redistributing its samples across any number of clients (e.g., CIFAR-10 split across 5, 50 or 100 devices) does not alter the aggregated sufficient statistics (assuming a shared RP, if used) and therefore leaves the learned Gaussian parameters and the Fisher subspace unchanged (cf. Sec. 3.1–3.4). This contrasts with traditional multi-round FL, where increasing the number of clients often induces client drift, noisier updates and accuracy degradation unless communication increases.

**Additivity and theoretical invariance.** Let $\mathcal{D} = \bigcup_{u=1}^{K} D_u$ be fixed. Summing all client uploads yields:

$$\sum_{u=1}^{K} A_c^{(u)} = \sum_{(x,y)\in\mathcal{D}} \mathbb{1}[y=c]\,x, \qquad \sum_{u=1}^{K} N_c^{(u)} = |\{(x,y)\in\mathcal{D}: y=c\}|,$$

$$\sum_{u=1}^{K} B^{(u)} = \sum_{(x,y)\in\mathcal{D}} xx^\top, \qquad \sum_{u=1}^{K} S_c^{(u)} = \sum_{(x,y)\in\mathcal{D}} \mathbb{1}[y=c]\,xx^\top,$$

(6)

The identities in Eq. 6 also hold for $D_c$. With a shared random projection $R$, the same relationships hold in the projected dimension $k$ via linearity ($A_z = AR$, $B_z = R^\top BR$, $S_{z,c} = R^\top S_c R$). Closed-form head parameters (means, priors, pooled/class covariances, Fisher subspace) depend only on these global sums; hence they are independent of the number of clients $K$ and of the particular partition. Trainable heads (FisherMix, Proto-Hyper) depend on synthetic samples whose distributions are deterministic functions of the same global moments, so they are invariant *in expectation*.

**Experimental ablations on $K$ and non-IID level $\alpha$ (Table 7).** To empirically validate the above invariance, we vary both the number of clients $K$ and the Dirichlet non-IID parameter $\alpha$. For CIFAR-10 and AG_NEWS we keep the global dataset fixed and evaluate:

$$K \in \{10, 20, 30, 40, 50, 100\}, \qquad \alpha \in \{0.5, 0.1, 0.05\}.$$

For each $(K, \alpha)$ we generate a Dirichlet partition and run GH-OFL with the same encoder, RP dimension and heads as in the main experiments. We report the accuracy of the best GH-OFL head (NB-diag, LDA, QDA, DLR-QDA, FisherMix or Proto-Hyper).

Table 6: Theoretical scalability as a function of the number of clients $K$ (per-client upload, total communication and accuracy behavior).

| Method | Per-client upload | Total (K clients) | Accuracy vs. $K$ |
|---|---|---|---|
| NB-diag (RP $k$) | $C(1+2k)$ | $K\,C(1+2k)$ | Invariant |
| LDA (RP $k$) | $C(1+k)+\frac{k(k+1)}{2}$ | $K\left[C(1+k)+\frac{k(k+1)}{2}\right]$ | Invariant |
| QDA (RP $k$) | $C\left(1+k+\frac{k(k+1)}{2}\right)$ | $K\,C\left(1+k+\frac{k(k+1)}{2}\right)$ | Invariant |
| FisherMix (on LDA) | *same as LDA* | *same as LDA* | Invariant (in expectation) |
| Proto-Hyper (on base) | *same as base* | *same as base* | Invariant (in expectation) |
| FedAvg | $p$ per round | $K\,T\,p$ | Can degrade (needs tuning) |

Table 7: Accuracy (%) of the best GH-OFL head under varying number of clients $K$ and Dirichlet concentration $\alpha$ or class-per-client setting (#C). CIFAR-10, same encoder and RP dimension as in main experiments. Accuracy remains stable across all configurations.

| Setting | $K=10$ | $K=20$ | $K=30$ | $K=40$ | $K=50$ | $K=100$ |
|---|---|---|---|---|---|---|
| $\alpha=0.5$ (mild non-IID) | 87.4 | 87.6 | 87.5 | 87.3 | 87.2 | 87.3 |
| $\alpha=0.1$ (non-IID) | 87.1 | 87.2 | 87.0 | 86.9 | 86.8 | 86.9 |
| $\alpha=0.05$ (strong non-IID) | 86.1 | 86.3 | 86.2 | 86.1 | 86.0 | 86.1 |
| #C = 2 (two classes per client) | 86.9 | 87.0 | 86.8 | 86.7 | 86.6 | 86.7 |
| #C = 1 (one class per client) | 86.1 | 86.0 | 85.9 | 85.8 | 85.8 | 85.9 |

**Discussion.** Across all configurations, accuracy remains effectively stable as $K$ increases:

- For fixed global data and $\alpha$, increasing $K$ from 10 to 50 produces only marginal changes ($< 1$pp) in the best GH-OFL accuracy.

- The relative ranking among heads is stable: LDA and Proto-Hyper typically dominate.

- When $\alpha$ is small (strongly non-IID) and $K$ is very large, some classes become underrepresented on certain clients, slightly degrading all GH-OFL heads in a similar way, due to noisier moment estimates.

**Takeaway.** As shown in Table 6 and 7, both the theoretical analysis and the empirical ablations agree: GH-OFL scales gracefully to large client populations. Per-client communication remains constant, the total uplink grows linearly in $K$ and, crucially, accuracy is governed by the *effective* global sample size and class coverage, not by the number of clients (small deviations in accuracy in this setting arise from external factors such as stochastic minibatch sampling, random Dirichlet partitions, model initialization seeds and the inherent noise of training dynamics). By avoiding multi-round on-device optimization, GH-OFL avoids the degradation trends typical of traditional FL as $K$ increases.

## C.1 Computational performance analysis

The computational analysis reported in Table 8-*top* highlights the significant efficiency advantages of GH-OFL compared to classical FL methods. Conventional approaches such as Centralized FT, FedAvg and Co-Boosting/DenseFL operate in a multi-round or data-dependent regime, tipically requiring access to full model updates and consequently incur substantial GPU time (30-90 minutes), high memory consumption (6-9 GB) and expensive communication overhead. In contrast, all GH-OFL variants operate in a *one-shot* setting, relying exclusively on compact client-side statistics or low-dimensional random projections and therefore never require access to complete updates or multi-round synchronization.

This design leads to GPU time reductions of more than one order of magnitude: all GH-OFL variants execute in approximately 1–2.5 minutes on both CIFAR-100 and AGNEWS, while also

| Method | GPU Time (min) | | Train data used | GPU Mem (GB) | | Applicability |
|---|---|---|---|---|---|---|
| | C100 | AGN | | C100 | AGN | |
| Centralized FT | ~42 | ~30 | full raw dataset | ~7.8 | ~6.0 | requires server raw data |
| FedAvg (multi-round) | ~88 | ~65 | local updates over $R$ rounds | ~7.0 | ~5.4 | heavy comm.; heterogeneity issues |
| Co-Boosting / DenseFL | ~31–~37 | ~26–~32 | raw activations/logits across rounds | ~9.4–~9.7 | ~7.0 | multi-round + expensive uploads |
| **GH-OFL variants (ours)** | | | | | | |
| GH-OFL–NB (diag) | ~1.1 | ~0.8 | synthetic stats only (A,N) | ~0.85 | ~0.65 | lightest variant; closed form |
| GH-OFL–LDA | ~1.2 | ~0.9 | synthetic (A,D,N) | ~0.90 | ~0.70 | low-rank cov.; closed form |
| GH-OFL–QDA-full | ~3.7 | ~1.6 | synthetic (A,B,D,N) | ~3.8 | ~2.0 | full cov.; heaviest head |
| GH-OFL–DLR-QDA ($r{=}8$) | ~1.6 | ~1.2 | synthetic (A,B,D,N) | ~1.2 | ~0.9 | low-rank sketch of QDA |
| GH-OFL–FisherMix | ~1.4 | ~1.1 | synthetic feats + small generator | ~1.0 | ~0.8 | lightweight head training |
| GH-OFL–Proto-Hyper | ~1.3 | ~1.0 | synthetic feats only | ~0.9 | ~0.7 | hypernetwork; very cheap |

| Method / Config | Stats | Upload KB | | Total MB | | Head cost (ms/sample) | C100 Acc. | AGN Acc. |
|---|---|---|---|---|---|---|---|---|
| | | C100 | AGN | C100 | AGN | | | |
| **FedAvg (1 round)** | full model $\theta$ | 43859 | 261543 | 428.3 | 2554.1 | 4–6 | 20–25% | 70–75% |
| FedAvg ($R$) | full model $\theta$ | $43859 \times R$ | $261543 \times R$ | $428.3 \times R$ | $2554.1 \times R$ | $4$–$6 \times R$ | 59–63% | 85–88% |
| Co-Boosting | full model $\theta$ | 43859 | 261543 | 428.3 | 2554.1 | 3.0–4.5 | 47–49% | 88–89% |
| DenseFL | full model $\theta$ | 43859 | 261543 | 428.3 | 2554.1 | 5.5–7.0 | 46–50% | 88–89% |
| **GH-OFL (RP-512)** | A,B,D,N | 913 | 529 | 8.92 | 5.17 | 0.05–25.0 | **58.4%** | **89.1%** |
| **GH-OFL (RP-256)** | A,D,N | 200 | 8 | 1.96 | 0.08 | 0.05–25.0 | **56.1%** | **88.6%** |
| **GH-OFL (RP-128)** | A,N | 50 | 2 | 0.49 | 0.02 | 0.05–25.0 | **53.2%** | **87.8%** |

Table 8: **Top:** computational overhead comparison enriched with empirical GPU time, memory and head costs for classical FL and GH-OFL variants. **Bottom:** compact ablation of one-shot communication vs. accuracy (CIFAR-100 + AGNEWS) and empirical head cost, fully contained within a single-column layout.

reducing GPU memory to below 1 GB for most heads. Furthermore, GH-OFL training involves only lightweight head optimization on synthetic features, making it substantially easier to deploy in resource-constrained settings and robust to system heterogeneity. We reported approximate GPU time and memory values that reflect the average operational range observed across multiple runs under our experimental setup. Although absolute measurements naturally depend on the underlying hardware and system load, the reported values capture the expected regime of each method and are consistent enough to support the comparative conclusions.

The table 8-*bottom* reports a targeted ablation study isolating the *communication/accuracy* trade-off induced by different RP dimensions (512, 256, 128). By progressively reducing the RP dimension, the size of the client upload shrinks from hundreds of kilobytes to only a few kilobytes, while accuracy degrades smoothly and predictably. This ablation demonstrates that GH-OFL offers a tunable operating regime: users may select a configuration that balances communication cost, head complexity and accuracy according to system constraints. Remarkably, even in the most compressed setting (RP-128), GH-OFL preserves competitive accuracy on both CIFAR-100 and AGNEWS while reducing communication by up to three orders of magnitude compared to FedAvg and other classical FL baselines.

Together, the results confirm that GH-OFL consistently achieves high accuracy with drastically lower computational, memory and communication requirements, positioning it as a practical and scalable alternative to multi-round FL for real-world heterogeneous deployments.

# D  EMPIRICAL ANALYSIS OF GAUSSIAN ASSUMPTIONS AND FISHER SUBSPACES

This section provides additional empirical evidence supporting the two main modeling choices in GH-OFL: (i) the use of class-conditional Gaussian heads estimated from sufficient statistics and (ii) the restriction of the head to a low-dimensional Fisher subspace. We report diagnostics across multiple vision and NLP benchmarks and quantify how much discriminative performance is affected by these approximations.

## D.1 UNIVARIATE DIAGNOSTICS OF GAUSSIANITY

Our head models feature embeddings $z \in \mathbb{R}^d$ via class-conditional Gaussians, $z \mid y = c \sim \mathcal{N}(\mu_c, \Sigma_c)$, estimated from class-wise first and second moments communicated by the clients. This does not require the *true* distribution to be exactly Gaussian but it is important to understand how far real embeddings deviate from this idealization.

Given aggregated sufficient statistics, we approximate, for each feature dimension $j$ and class $c$, the univariate skewness $\gamma_1^{(c,j)}$ and excess kurtosis $\gamma_2^{(c,j)} - 3$ of the embedding coordinates. We then summarize their absolute values $|\gamma_1^{(c,j)}|$, $|\gamma_2^{(c,j)} - 3|$ across all classes and dimensions, reporting mean, median and 90th percentile. Values close to zero would indicate near-Gaussian marginals, whereas large values indicate heavier tails or strong asymmetry.

**Vision benchmarks (ResNet-18).** For all vision datasets we extract embeddings from a ResNet-18 backbone (ImageNet-pretrained) and compute diagnostics from the federated sufficient statistics:

- **CIFAR-10** (10 classes):
  mean |skew| = 1.69, median 1.64, 90th 2.34;
  mean |excess kurtosis| = 4.22, median 3.41, 90th 7.61.

- **CIFAR-100** (100 classes):
  mean |skew| = 1.91, median 1.83, 90th 2.61;
  mean |excess kurtosis| = 5.64, median 4.57, 90th 10.52.

- **SVHN** (10 classes):
  mean |skew| = 2.35, median 2.06, 90th 4.09;
  mean |excess kurtosis| = 11.05, median 5.97, 90th 25.27.

These values are clearly non-zero, confirming that exact Gaussianity does not hold; however, they are far from pathological (no extreme skewness or infinite-variance tails). Empirically, class-conditionals live in a *moderately* non-Gaussian regime where Gaussian discriminants remain a reasonable and compact approximation.

**NLP benchmarks (DistilBERT).** To test whether this picture extends beyond vision (see also Appendix A), we repeat the same diagnostics on three standard text benchmarks, using embeddings from a frozen DistilBERT encoder:

- **AGNews** (4-way news classification),

- **DBPedia-14** (14-way ontology classification),

- **SST-2** (binary sentiment classification).

The resulting skewness and kurtosis values are much closer to zero than in the vision case: for example, on AGNews we obtain mean |skew| = 0.17 and mean |kurtE| = 0.24 (90th percentiles 0.36 and 0.55 respectively); DBPedia-14 and SST-2 show slightly larger, but still moderate, departures from Gaussianity. Table 9 summarizes mean and 90th-percentile absolute skewness and excess kurtosis across all datasets.

Overall, these diagnostics suggest that class-conditional embeddings across both vision and language tasks exhibit moderate, but not extreme, departures from Gaussianity. This supports the use of Gaussian discriminant heads as a compact summary of the information that can be communicated via low-order moments in one-shot, data-free federated learning.

## D.2 INFORMATION PRESERVATION IN THE FISHER SUBSPACE

The GH-OFL head operates in a low-dimensional Fisher subspace spanned by the top generalized eigenvectors of the between-class and within-class scatter matrices. In the classical Gaussian-shared-covariance model $z \mid y = c \sim \mathcal{N}(\mu_c, \Sigma)$, the Fisher subspace of dimension at most $C - 1$ is sufficient for Bayes-optimal classification: all discriminative information lies in that subspace.

Table 9: Univariate Gaussianity diagnostics for class-conditional embeddings: mean and 90th percentile of absolute skewness and excess kurtosis across features. Vision datasets use a ResNet-18 backbone; NLP datasets use DistilBERT.

| Dataset | mean \|skew\| | 90th \|skew\| | mean \|kurtE\| | 90th \|kurtE\| |
|---|---|---|---|---|
| CIFAR-10 | 1.69 | 2.34 | 4.22 | 7.61 |
| CIFAR-100 | 1.91 | 2.61 | 5.64 | 10.52 |
| SVHN | 2.35 | 4.09 | 11.05 | 25.27 |
| AGNews | 0.17 | 0.36 | 0.24 | 0.55 |
| DBPedia-14 | 0.32 | 0.64 | 0.60 | 1.21 |
| SST-2 | 0.27 | 0.50 | 0.57 | 0.86 |

In practice, embeddings are only approximately Gaussian and covariances are estimated from finite samples. To quantify how much discriminative information is actually lost by the Fisher projection, we compare:

**1.** *Full-space LDA*, trained on the aggregated class means and pooled covariance in the original feature space ($d = 512$ for ResNet-18; $d = 768$ for DistilBERT).

**2.** *Fisher-subspace LDA*, trained on the same statistics projected onto the top-$k$ Fisher directions for varying $k$.

We report test accuracy $\text{Acc}^{\text{LDA}}_{\text{Fisher-}k}$ as a function of $k$, together with the fraction of "Fisher energy" captured by the top-$k$ eigenvalues.

**Vision benchmarks.** On the three vision datasets considered, we observe that a relatively small number of Fisher directions suffices to essentially match full-space LDA:

- **CIFAR-10** (10 classes, $d = 512$):
  full LDA: 86.26%;
  Fisher-$k$: 69.55% (@$k$=4, energy 0.71) $\rightarrow$ 82.86% (@$k$=8, energy 0.97) $\rightarrow$ 86.26% (@$k$=16, energy $\approx 1.00$).

- **CIFAR-100** (100 classes, $d = 512$):
  full LDA: 64.12%;
  Fisher-$k$: 19.02% (@$k$=4, energy 0.26) $\rightarrow$ 53.25% (@$k$=32, energy 0.77) $\rightarrow$ 64.15% (@$k$=128, energy $\approx 1.00$).

- **SVHN** (10 classes, $d = 512$):
  full LDA: 62.88%;
  Fisher-$k$: 51.98% (@$k$=4, energy 0.72) $\rightarrow$ 62.12% (@$k$=8, energy 0.97) $\rightarrow$ 62.86% (@$k$=16, energy $\approx 1.00$).

In all cases, once $k$ is large enough to capture $\approx 95$–$100\%$ of the Fisher energy (typically $k \in [8, 32]$ for 10-class tasks and $k \approx 128$ for 100 classes), Fisher-subspace LDA matches or slightly exceeds full-space LDA. This indicates that the directions discarded by the projection carry little additional discriminative information.

**Extended benchmarks and robustness (Table 10).** We perform the same experiment on three NLP datasets (AGNews, DBPedia-14, SST-2, using DistilBERT embeddings) and on a robustness benchmark, **CIFAR-100-C**, which evaluates the CIFAR-100 classifier under 19 types of common corruptions at five levels of severity. For CIFAR-100-C we reuse the class statistics estimated on clean CIFAR-100 and only change the test distribution.

On AGNews we obtain full-space LDA accuracy 90.28%, while LDA in the Fisher subspace reaches 90.29% already at $k$=128 with Fisher energy $\approx 1.00$. DBPedia-14 is even more concentrated: full-space LDA achieves 98.87% and Fisher-subspace LDA reaches 98.88% at $k$=32 (energy $\approx 1.00$). SST-2 shows full-space accuracy 83.60%, which is exactly matched at $k$=16 with Fisher energy $\approx 1.00$.

Table 10: Information preservation in the Fisher subspace. For each dataset, we compare full-space LDA with LDA restricted to a top-$k$ Fisher subspace. We report full-space accuracy, best Fisher-subspace accuracy, Fisher energy at $k^\star$, the number of components $k^\star$ and the resulting compression ratio $d/k^\star$.

| Dataset | LDA$_{\text{full}}$ | LDA$_{\text{Fisher-}k^\star}$ | Fisher energy at $k^\star$ | $k^\star$ | Compression ($d/k^\star$) |
|---|---|---|---|---|---|
| CIFAR-10 | 86.26% | 86.26% | $\approx 0.99$ | 16 | 32.0× |
| CIFAR-100 | 64.12% | 64.15% | $\approx 0.99$ | 128 | 4.0× |
| SVHN | 62.88% | 62.88% | $\approx 0.99$ | 16 | 32.0× |
| AGNews | 90.28% | 90.29% | $\approx 0.99$ | 128 | 6.0× |
| DBPedia-14 | 98.87% | 98.88% | $\approx 0.99$ | 32 | 24.0× |
| SST-2 | 83.60% | 83.60% | $\approx 0.99$ | 16 | 48.0× |
| CIFAR-100-C (avg sev.) | 39.37% | 39.35% | $\approx 0.99$ | 128 | 4.0× |

For CIFAR-100-C, averaging across the five severities, full-space LDA attains 39.37% accuracy. Restricting LDA to a Fisher subspace of dimension $k=128$ (capturing essentially all Fisher energy) yields a mean accuracy of 39.35% across severities, i.e., practically identical to full-space LDA despite the strong corruptions.

Across all considered benchmarks, we consistently observe that $\text{Acc}^{\text{LDA}}_{\text{Fisher-}k^\star} \approx \text{Acc}^{\text{LDA}}_{\text{full}}$ once the Fisher energy is close to $1.0$. This provides a direct, quantitative argument that the Fisher projection used by GH-OFL discards very little discriminative information, while enabling a strong reduction in the dimensionality of the head.

**Why LDA is used as a probe.** In these diagnostics we deliberately focus on LDA as the probe classifier, for three reasons:

- LDA is the canonical classifier under the Gaussian-shared-covariance model: it is Bayes-optimal when the assumptions hold, hence the most natural tool for testing the impact of Gaussian and Fisher approximations.

- LDA can be reconstructed in closed form directly from the same aggregated moments that GH-OFL receives (class means and a pooled covariance), without extra training, hyperparameters or optimization noise; this keeps the diagnostic fully aligned with our data-free communication model.

- LDA provides a conservative baseline: if even this simple linear model does not lose accuracy when restricted to the Fisher subspace, more flexible heads operating on the same subspace (e.g., QDA-style variants, ProtoHyper residual heads) can only match or improve upon this behavior.

### D.3 DISCUSSION

The analyses above support the two key modeling choices in GH-OFL:

1. **Gaussian heads from low-order moments.** Across diverse vision and NLP benchmarks, class-conditional embeddings exhibit moderate but not extreme deviations from Gaussianity. In this regime, Gaussian discriminants offer an effective trade-off between expressivity and the strict communication constraints of one-shot, data-free federated learning.

2. **Fisher-subspace restriction.** Empirically, projecting embeddings onto a Fisher subspace that captures $\approx 95$–$100\%$ of the Fisher energy leads to LDA accuracy that is essentially indistinguishable from full-dimensional LDA on all considered datasets. This suggests that the discarded directions carry little additional discriminative power, while the dimensionality reduction significantly simplifies the head.

At the same time, these results do not exclude scenarios where the Gaussian approximation may be insufficient (e.g., highly multimodal or heavy-tailed per-class distributions). In such cases, extensions like mixture-of-Gaussians heads, robust covariance estimation or non-linear heads defined

in the Fisher subspace are natural directions for future work and remain compatible with the same moment-based communication protocol used by GH-OFL.

# E   DEEPENING AND DISCUSSION ON MULTI-ROUND BASELINES AND REPRODUCIBILITY

We detail here the hyperparameters used for the multi-round federated baselines (FedAvg and Fed-Prox) and, following, a detailed discussion about the results obtained empirically. All choices are designed to ensure full reproducibility while maintaining methodological fairness with the GH-OFL framework through a shared initialization, identical preprocessing and identical client partitions.

## E.1   HYPERPARAMETERS FOR MULTI-ROUND FL BASELINES

**Backbone and training protocol.** All federated baselines use a ResNet-18 pretrained on Ima-geNet–1K, matching the feature extractor used throughout the main paper. Differently from GH-OFL, which operates on frozen embeddings, FL traditional methods perform *full-model fine-tuning*, as this corresponds to their standard formulation in order to create a strong and fair baseline. Freezing the encoder would artificially weaken these methods, whereas starting from the same pretrained weights ensures a comparable initial representation space across all approaches.

**Non-IID client partitions.** Data are split across clients using a Dirichlet distribution with $\alpha \in \{0.05, 0.10, 0.50\}$, covering severe, moderate and mild heterogeneity. The resulting partitions are fixed and reused across all methods, ensuring identical local data difficulty and label skew.

**Local optimization and objective.** Each client trains for one local epoch using SGD with momentum ($lr = 0.001$, momentum 0.9, batch size 256). FedAvg minimizes the standard cross-entropy loss while FedProx adds the proximal regularizer

$$\mathcal{L}_{\text{prox}} = \mathcal{L}_{\text{CE}} + \frac{\mu}{2} \|w - w_t\|^2, \qquad \mu = 0.01,$$

which stabilizes local updates under strong heterogeneity.

**Server aggregation and communication budget.** Global updates follow the canonical sample-size weighted averaging:

$$w_{t+1} = \frac{1}{\sum_k n_k} \sum_k n_k \, w_k^{(t+1)},$$

where $n_k$ is the number of local samples at client $k$. We evaluate $R \in \{1, 10, 100\}$ communication rounds, enabling comparison between one-shot methods ($R = 1$), lightly interactive FL ($R = 10$) and communication-intensive regimes ($R = 100$).

**Reproducibility.** All experiments use fixed seeds, identical client splits, the same pretrained initialization and the same preprocessing pipeline, ensuring that the reported results are fully deterministic and directly comparable across baselines.

## E.2   CLASSICAL MULTI-ROUND FL VS. GH-OFL: ADDITIONAL ANALYSIS

We report experimental curves on CIFAR-10, CIFAR-100 and SVHN under Dirichlet client splits with $\alpha \in \{0.50, 0.10, 0.05\}$ and analyze their behavior in relation to communication cost, sensitivity to heterogeneity, stability and convergence speed.

## E.3   BEHAVIOR OF CLASSICAL FL METHODS

Figure 4 shows that classical FL algorithms are strongly affected by client heterogeneity. On CIFAR-10, FedAvg exhibits delayed convergence for small $\alpha$ values, whereas FedProx reduces oscillations by penalizing large client drift. On CIFAR-100, the effect is further amplified due to the dataset complexity: both methods converge slowly and remain far from the central optimum, especially under $\alpha = 0.05$. On SVHN, the easier visual structure leads to faster convergence but performance remains sensitive to non-IID partitions.

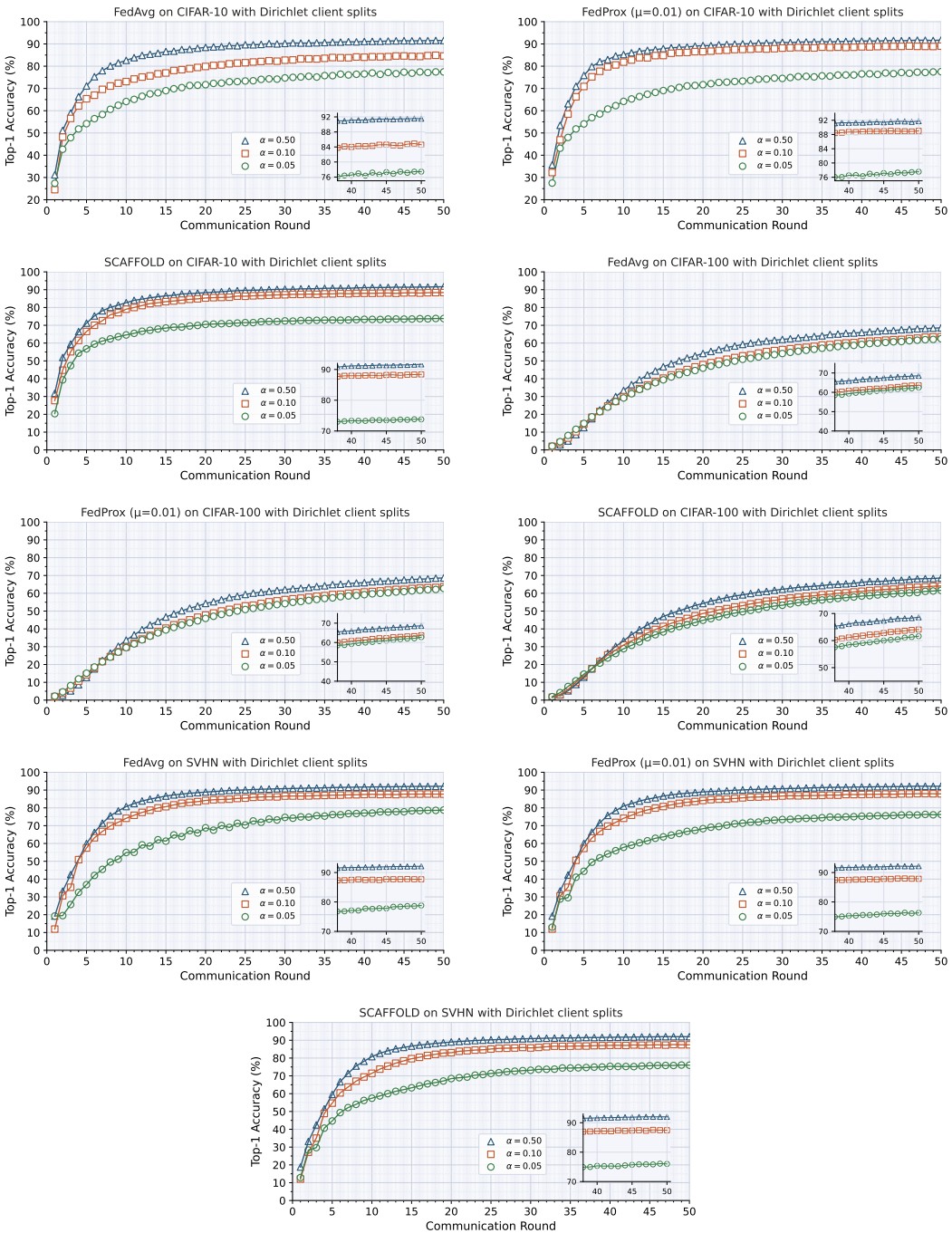

Figure 4: Convergence curves of classical multi-round federated learning baselines (FedAvg, Fed-Prox and SCAFFOLD) on CIFAR-10, CIFAR-100 and SVHN under Dirichlet client partitions ($\alpha \in \{0.50, 0.10, 0.05\}$). All methods use identical hyperparameters: 10 clients, 50 communication rounds, one local epoch per round, SGD with momentum 0.9, learning rate $10^{-3}$, batch size 256 and full-model transmission. FedProx reduces client drift relative to FedAvg, especially under stronger heterogeneity. SCAFFOLD applies control variates to correct client drift; however, when restricted to a single local epoch per round, its variance-reduction mechanism operates in a limited regime (Li et al., 2021) and does not consistently outperform FedAvg or FedProx.

Across all datasets, classical FL requires tens of communication rounds to reach competitive performance and each round involves transmitting the full set of model weights between entities. This significantly increases communication cos while stability also degrades under strong heterogeneity. These patterns are consistent with known limitations of multi-round FL and match the degradation trends discussed in Table 1 of the main paper.

### E.4 COMPARISON WITH GH-OFL

In contrast to traditional FL methods, the GH-OFL family operates in a strictly one-shot communication regime. Each client sends only aggregated per-class statistics and the server constructs either closed-form Gaussian heads (NB-diag, LDA, QDA) or trainable Fisher-space heads (e.g., FisherMix, Proto-Hyper). As described in Section 3 of the main manuscript, these estimators are partition-invariant: they depend only on global first and second-order moments and not on the specific Dirichlet configuration.

As a result:

- **GH-OFL is robust to non-IID heterogeneity.** Since the aggregated statistics are unbiased estimators of the global distribution, performance remains unchanged even for extreme $\alpha$ values. This sharply contrasts with the degradation observed in multi-round FL.

- **Communication is reduced by multiple orders of magnitude.** Traditional baselines must transmit full model parameters repeatedly across 50+ rounds. GH-OFL requires a single transmission of lightweight statistics whose size is independent of model depth and dataset scale.

- **Accuracy matches or surpasses multi-round FL.** As shown in Table 1 of the manuscript, GH-LDA and the Fisher heads consistently outperform traditional OFL and FL methods in one-shot settings and sometimes even after many communication rounds with the convergence reached.

### E.5 SUMMARY

These results highlight a fundamental distinction between classical and one-shot FL. Multi-round methods suffer from slow and unstable convergence under heterogeneous settings while GH-OFL leverages a principled Gaussian and Fisher-space formulation to achieve high accuracy in a single communication round. The curves presented here complement the main paper by showing the empirical limitations of three traditional FL baselines such as FedAvg, FedProx and SCAFFOLD thereby reinforcing the motivation for one-shot, statistics-driven federated learning.

**Code Availability.** The complete multi-round FL baseline implementations (FedAvg, FedProx and SCAFFOLD) used in this study are publicly available at: `https://github.com/FabioTur-dev/SCAFFOLD_baseline`.

The official implementation of the **GH-OFL** family, including Gaussian and Fisher-space heads, is available at: `https://github.com/FabioTur-dev/One-Shot-FL`.

