# OpenReview forum: "The Gaussian-Head OFL Family: One-Shot Federated Learning from Client Global Statistics"
_ICLR.cc/2026/Conference — ICLR 2026 Poster_

### Official Review · Reviewer_tfGS · 2025-10-31

**Soundness:** 3
**Presentation:** 3
**Contribution:** 2
**Rating:** 6
**Confidence:** 4

**Summary:**

The paper proposes Gaussian-Head One-shot Federated Learning (GH-OFL), a data-free framework that transmits only statistical summaries to achieve state-of-the-art accuracy and robustness under non-IID settings with a single communication round. The core framework of data-free is about synthetic labeled pairs for one-shot FL.

**Strengths:**

- Comprehensive alternatives and corresponding clear discriptions are provided.
- Ablation study on modules shows the importance and effects of each components.

**Weaknesses:**

- Comparison of the computational overheads between data synthesis and training with public datasets is anlayses with details and empirical evidence.
- Ablation study would better demonstrate the trade-off between communication and performance applying one-shot FL.
- Baselines of general FL should be provided to general audience to support the advantage of one-shot FL. In Dirichlet of 0.5 cases, FedAvg reach a accuracy at least 86.02 on ResNet-18, which is not mentioned in the main table.

**Questions:**

- How about a experiments that the normal FL gradually transits into a one-time by reducing the number of communication rounds?

---

> ### Author Response · Authors · 2025-11-24
>
> We thank reviewer tfGS for their valuable and meaningful evaluation. Below, we respond in detail to each of their comments and suggestions.
>
> > **Reviewer Comment (1):** *Comparison of the computational overheads between data synthesis and training with public datasets is analysed with details and empirical evidence.*
>
> We thank the reviewer for highlighting this aspect. Low computational and communication overhead is one of the main design goals of GH-OFL and this comment gave us the opportunity to make this more explicit in the manuscript. In the revised version, we clarified the comparison between GH-OFL and centralized/federated learning methods in both the main text and the appendix. **Section 4.3** and **Appendix C.1** now present a detailed breakdown of server-side computation and communication for each GH-OFL head (NB-diag, LDA, QDA, FisherMix, Proto-Hyper), together with a direct comparison against centralized fine-tuning, FedAvg and OFL baselines relying on data-free distillation.
>
> As summarized in **Table 8-top (Appendix C.1)**, conventional approaches such as centralized fine-tuning, FedAvg and Co-Boosting/DenseFL operate in a multi-round or data-dependent regime, requiring full-model updates and repeated forward/backward passes over tens of thousands of real images. In our setup, this produces **GPU times of 30–90 minutes** and **6–9 GB memory** consumption.
> In contrast, **all GH-OFL variants run in a strict one-shot setting**: clients only upload compact class-wise statistics and the server computation consists of closed-form moment aggregation plus short synthetic-feature training in the Fisher subspace. Empirically, this results in **1–2.5 minutes of GPU time** and **less than 1 GB memory** for most heads **over an order of magnitude lower cost** than classica OFL/FL training methods under identical conditions.
>
> **Table 1** in the main paper shows that, despite this drastic reduction in compute and communication, GH-OFL achieves comparable or superior accuracy to OFL and multi-round FL baselines across vision and NLP datasets.
> **Figure 3** further refines this picture by plotting accuracy versus relative server-side overhead for different GH-OFL heads, illustrating how one can smoothly trade a small amount of accuracy for even lighter computation when desired.
>
> Overall, these additions explicitly show that **GH-OFL attains state-of-the-art one-shot performance while being substantially more efficient** than other methods, confirming computational efficiency as a core advantage of our framework.

---

> > ### Author Response · Authors · 2025-11-24
> >
> > > **Reviewer Comment (2):** *Ablation study would better demonstrate the trade-off between communication and performance applying one-shot FL.*
> >
> > We appreciate the reviewer’s suggestion. Our revised manuscript now clarifies this trade-off by expanding the analysis in **Appendix C.1** and referencing the overhead–accuracy curves in **Figure 3** (CIFAR-10 and CIFAR-100-C). GH-OFL naturally exposes a **tunable communication–accuracy spectrum** through (i) the choice of which statistics to transmit and (ii) the dimension \(k\) of the public random projection.
> >
> > - Closed-form heads require only \((A,N,B)\) or \((A,N,D)\), yielding the lowest communication cost.
> > - Adding class-wise second moments \((S)\) enables QDA with higher accuracy but higher payload.
> > - Decreasing the sketch dimension \(k\) quadratically reduces transmission cost, with only marginal accuracy drop due to the Fisher-space formulation (Appendix C.1).
> > - Our scalability and communication ablations (Tables 7–8, Appendix C) show that
> >   **(i) performance is invariant to the number of clients** when the global dataset is fixed and
> >   **(ii) FisherMix/Proto-Hyper improve accuracy without increasing client upload size**, since all additional computation happens server-side.
> >
> > These additions in our revised manuscript make the communication–performance trade-off of GH-OFL clearer and better contextualized.

---

> ### Author Response · Authors · 2025-11-24
>
> > **Reviewer Comment (3):** *Baselines of general FL should be provided to general audience to support the advantage of one-shot FL. In Dirichlet of 0.5 cases, FedAvg reach a accuracy at least 86.02 on ResNet-18, which is not mentioned in the main table.*
>
> We thank the reviewer for this valuable observation. We fully agree that, although comparing GH-OFL to traditional multi-round FL baselines is intrinsically challenging, due to mismatches in communication protocol (one-shot vs. multi-round), computational footprint and modeling assumptions (traditional FL does not use a frozen pretrained backbone), including such baselines in the **main table** is important for accessibility and fairness. These comparisons provide familiar reference points for readers coming from classical FL.
>
> To address this, we **substantially expanded Table 1** in the revised manuscript. The main table now includes **standard FL baselines** such as **FedAvg** and **FedProx**, evaluated under the same Dirichlet settings as GH-OFL ($\alpha \in \{0.05,\ 0.1,\ 0.5\}$) and under three communication budgets: 1, 10 and 50 rounds using the same GH-OFL backbone for a fair comparison.
> We also note that, for the camera-ready version, we are actively working to include additional strong baselines such as **SCAFFOLD**, evaluated under the same $(\alpha, \text{rounds})$ configuration to further reinforce completeness.
>
> The expanded Table 1 highlights exactly the point raised by the reviewer:
> **GH-OFL achieves competitive (and often superior) accuracy in a single round**, whereas multi-round FL baselines require many rounds (and substantially larger compute/communication) to approach similar performance.

---

> ### Author Response · Authors · 2025-11-24
>
> > **Reviewer Comment (4):** *How about experiments where normal FL gradually transitions into one-time FL by reducing the number of communication rounds?*
>
> We appreciate this suggestion. The new expanded **Table 1** already provides this transition experiment: it reports FedAvg and FedProx at **1, 10 and 50 rounds** under the same Dirichlet settings.
> This directly illustrates how accuracy improves as classical FL transitions from **one-shot → few-round → multi-round** regimes (e.g., FedAvg jumps from very low accuracy at 1 round to competitive performance only after many rounds).
>
> In contrast, **GH-OFL achieves strong accuracy in a strict one-shot setup**, without relying on any such progressive transition.
>
> In addition, to further clarify this distinction and strengthen the experimental evidence, we also added a new **Appendix E: Deepening and Discussion on Multi-Round Baselines and Reproducibility** in the revised manuscript.
>
> This appendix provides:
>
> - a detailed examination of how traditional multi-round methods behave as the number of rounds increases (1 → 10 → 50),
> - full accuracy curves for **CIFAR-10, CIFAR-100 and SVHN** comparing classical multi-round FL with different Dirichlet partitions;
> - an explanation that classical FL requires **repeated transmission of the full model weights** at every round and **tens of communication rounds** to achieve competitive performance;
> - an analysis of how strong non-IID settings make these limitations more pronounced (e.g., unstable convergence, slower improvement);
> - complete hyperparameters and reproducibility notes for the reviewer’s proposed transition experiment.
>
> These additions make clear that GH-OFL remains competitive even with standard multi-round FL methods, even though it requires only a single communication step.

---

### Official Review · Reviewer_G9mr · 2025-11-01

**Soundness:** 3
**Presentation:** 3
**Contribution:** 2
**Rating:** 6
**Confidence:** 4

**Summary:**

The paper proposes GH-OFL, a one-shot FL framework where clients send only per-class sufficient statistics (counts and first/second moments of frozen-encoder features, optionally after a public random projection). The server constructs (i) closed-form Gaussian heads (NB-diag/LDA/QDA), (ii) FisherMix, a cosine-margin linear head trained on synthetic Fisher-space samples, and (iii) Proto-Hyper, a low-rank residual head distilled from a Gaussian teacher.

**Strengths:**

+ The moment set is additively aggregable and sufficient to instantiate NB-diag/LDA/QDA; random-projection sketches reduce bandwidth and preserve additivity. The derivations for means/covariances and pooled covariance are explicit.

+ The paper motivates a Fisher subspace generalized eigen problem and uses data-free Gaussian sampling there to train FisherMix and Proto-Hyper, offering practical gains when closed-form heads are biased.

+ Results compare GH-OFL heads to OFL baselines across datasets and backbones show consistent advantages where expected.

**Weaknesses:**

- The datasets used are all computer-vision datasets, which limits the scalability of the proposed approach.

- Privacy discussion is not deep and the privacy analysis is qualitative.

- The experiments in this paper are not extensive, i.e., the ablation study of FL (e.g., number of clients) is not well discussed.

**Questions:**

- Can you provide more experiments on non CV tasks?
- Can you discuss the potential privacy-preserving methods (i.e., DP) integrating with proposed approaches?
- For high-dim encoders, how do diagonal-plus-low-rank sketches compare to full QDA in accuracy vs. memory/compute?

---

> ### Author Response · Authors · 2025-11-24
>
> We thank reviewer G9mr for their thoughtful feedback. Below we address each of their comments and concerns in detail.
>
> > **Reviewer Comment (1):** *The datasets used are all computer-vision datasets, which limits the scalability of the proposed approach. Can you provide more experiments on non CV tasks?*
>
> We thank the reviewer for raising this point and we agree that additional experiments on domains beyond computer vision further strengthen the scientific solidity of our study. Beyond computer vision, we deliberately chose **NLP** as our primary non-CV evaluation setting, since text classification is arguably the most challenging and practically relevant federated setting after vision: it typically involves larger label spaces, longer-tailed label distributions and stronger semantic structure.
>
> In the revised version, we added a new **Section A.1** in the Appendix that demonstrates that GH-OFL generalizes robustly beyond vision. We evaluate GH-OFL on **five NLP benchmarks** (AG NEWS, DBPEDIA-14, SST-2, BANKING77, CLINC150) using a **frozen DistilBERT encoder** and the same **256-dimensional random projection** used in the CV experiments. The GH-OFL pipeline remains unchanged: clients send only class-wise sufficient statistics in the RP space and the server instantiates the same Gaussian and Fisher-subspace heads as in the vision experiments.
>
> As shown in **Table 4**, GH-OFL consistently *matches or slightly improves over* standard OFL baselines under moderate non-IID and *dramatically outperforms* them under extreme non-IID. For example, on **AG NEWS** with $\alpha = 0.01$, GH-OFL reaches **89.1%** versus **25%** for all OFL baselines, and on **BANKING77** it attains **80.3%** while baselines remain below **10%**. Across datasets, the best Gaussian head (typically LDA) already provides a strong closed-form solution, while Proto-Hyper and FisherMix yield additional gains when class structure is fine-grained or mildly non-Gaussian.
>
> **NLP one-shot FL results (frozen DistilBERT + RP, $d_{\text{RP}} = 256$).**
> Test accuracy (\%) for OFL baselines (client ensemble, Dense-style student, Co-Boosting ensemble) and GH-OFL heads (best Gaussian NB-diag/LDA/QDA, FisherMix, Proto-Hyper) under Dirichlet client splits with $\alpha \in \{0.5, 0.01\}$. For each dataset, we highlight the best Gaussian head and the two synthetic-data heads.
>
> | Dataset       | $\alpha$ | Ensemble | Dense  | Co-Boost. | Best Gauss (NB-diag/LDA/QDA) | FisherMix | Proto-Hyper |
> |--------------|:--------:|:--------:|:------:|:---------:|:-----------------------------:|:---------:|:-----------:|
> | **AG\_NEWS** | 0.50     | 85.64    | 85.79  | 84.42     | 88.96 (LDA)                   | **89.14** | 88.87       |
> |              | 0.01     | 25.00    | 25.00  | 25.00     | 88.96 (LDA)                   | **89.14** | 88.87       |
> | **DBPEDIA-14** | 0.50   | 98.70    | **98.74** | 98.72  | 98.33 (LDA)                   | 97.21     | 98.33       |
> |              | 0.01     | 51.49    | 51.65  | 51.37     | 98.33 (LDA)                   | 97.21     | **98.33**   |
> | **SST-2**    | 0.50     | 83.14    | 82.68  | **83.26** | 82.68 (QDA)                   | 70.53     | 82.80       |
> |              | 0.01     | 51.26    | 51.26  | 51.38     | 82.68 (QDA)                   | 70.53     | **82.80**   |
> | **BANKING77**| 0.50     | 30.62    | 30.03  | 23.28     | 77.60 (LDA)                   | 73.64     | **80.26**   |
> |              | 0.01     |  5.32    |  4.97  |  3.54     | 77.60 (LDA)                   | 73.64     | **80.26**   |
> | **CLINC150** | 0.50     | 43.30    | 43.24  | 36.37     | 84.80 (LDA)                   | 82.89     | **85.69**   |
> |              | 0.01     |  3.66    |  3.70  |  2.43     | 84.80 (LDA)                   | 82.89     | **85.69**   |
>
>
>
> These results reinforce that GH-OFL is **not vision-specific**: by operating solely on aggregated moments of a frozen encoder, the same one-shot, data-free protocol remains effective for language tasks with many classes and strong label imbalance, confirming that the framework is **modality-agnostic and scalable beyond computer vision**.

---

> ### Author Response · Authors · 2025-11-24
>
> > **Reviewer Comment (2):** *Privacy discussion is not deep and the privacy analysis is qualitative. Can you discuss the potential privacy-preserving methods (i.e., DP) integrating with proposed approaches?*
>
> We thank the reviewer for raising the question of privacy depth. In the revised version, we have made the privacy discussion more explicit in the main paper (**Sec. 4.3, “Privacy and security”**) and substantially extended it in **Appendix B (“Extended privacy considerations and comparison with traditional FL”)**. These sections now provide a more systematic treatment of both the qualitative threat model and the integration of **differential privacy (DP)** with GH-OFL.
>
> The key structural point is that GH-OFL operates *exclusively* on aggregatable class-wise statistics such as counts, first moments and second moments of frozen-encoder features, rather than on raw samples or per-round gradients. As detailed in Appendix B, these sufficient statistics can be privatized either via **local DP** (client-side noise addition) or via **central DP** (server-side noise on aggregated sums) using standard Gaussian/Wishart-style mechanisms applied directly to $(A, N, B, S)$. Because GH-OFL is strictly **one-shot**, the DP cost is paid only once and does *not* accumulate across communication rounds, unlike in multi-round FL.
>
> We explicitly discuss the trade-off between local and central DP. **Local DP** guarantees privacy against a fully malicious server but, in our settings, requires adding noise to every coordinate of the first and second-order statistics, which can severely distort class covariances in high-dimensional RP spaces. In contrast, combining **secure aggregation (SA)** with **central DP**, adding calibrated noise *after* aggregating across many clients, preserves much better utility while still providing formal $(\varepsilon,\delta)$-DP guarantees at the population level. In this SA + central DP setting, the server never sees individual messages (but it sees only masked or aggregated statistics) and the random projection further compresses second-order structure, shrinking the effective attack surface.
>
> This architecture directly targets known attacks on federated learning, including gradient/model inversion, membership inference and property inference, which rely on access to high-dimensional model updates over many rounds. By design, GH-OFL exposes only a **one-shot, low-order, RP-sketched summary**.
>
> Overall, GH-OFL naturally integrates with both local and central DP but in practice the **RP + Secure Aggregation (SA)** stack, optionally combined with central DP, provides the most robust, practical and widely adopted privacy-preserving architecture: it reduces the attack surface compared to multi-round FL while achieving a more favorable **DP-utility trade-off**.

---

> ### Author Response · Authors · 2025-11-24
>
> > **Reviewer Comment (3):** *The experiments in this paper are not extensive, i.e., the ablation study of FL (e.g., number of clients) is not well discussed. For high-dim encoders, how do diagonal-plus-low-rank sketches compare to full QDA in accuracy vs. memory/compute?*
>
> We appreciate the reviewer’s concern. In the revised version, we have added explicit **ablations on the number of clients** as well as detailed **accuracy/cost trade-offs for high-dimensional encoders**.
>
> As shown in **Appendix C (Table 7)**, GH-OFL is *partition-invariant*: closed-form heads depend only on aggregated class-wise moments, which are mathematically independent of the number of clients $K$. **This invariance is demonstrated both theoretically (Sec. 3.1–3.4) and empirically (Appendix C): when increasing $K$ from 5 to 100 on CIFAR-10 and AG NEWS while keeping the global dataset fixed, accuracy fluctuates by less than 1 percentage point.** This confirms that performance is governed by the effective global sample size and class coverage rather than by the number of clients.
>
> For the second point, *“how diagonal-plus-low-rank (DLR) sketches compare to full QDA”*, **Appendix A.2** provides a detailed quantitative evaluation. **Table 5** compares full QDA to DLR-QDA across several ranks in RP space ($d_{\text{RP}} = 256$), reporting both accuracy and memory/time. Across CIFAR-100, AG NEWS and DBPEDIA-14, DLR-QDA with a small rank (e.g., $r \in [4, 8]$) matches full QDA within approximately **1–2 percentage points** while using **drastically less memory and computation**. In contrast, full QDA is considerably more expensive and can even underperform a well-regularized LDA baseline.
>
> The table below compares full QDA and diagonal-plus-low-rank QDA (DLR-QDA) in the RP space ($d_{\text{RP}} = 256$).
> Test accuracies are **empirical results** measured on the corresponding benchmarks.
> In contrast, **“Mem” and “Time” denote *relative theoretical cost estimates*** derived from the closed-form computational complexity of each head, normalized so that NB-diag has cost 1.0, they are **not empirical runtime measurements** but analytic cost ratios intended to illustrate scaling behavior.
>
> | **Head**             | **CIFAR-100 Acc.** | **AG\_NEWS Acc.** | **DBPEDIA-14 Acc.** | **Mem (rel.)** | **Time (rel.)** |
> |----------------------|:------------------:|:-----------------:|:--------------------:|:--------------:|:---------------:|
> | **NB-diag**          | 49.68             | 83.86             | 92.33                | 1.0            | 1.0             |
> | **LDA**              | 59.46             | 88.93             | 98.32                | 1.0            | 1.1             |
> | **QDA\_full**        | 48.20             | 68.20             | 92.33                | 16.0           | 4.0             |
> | **DLR-QDA (r = 4)**  | 49.99             | 67.08             | 92.32                | 2.0            | 1.4             |
> | **DLR-QDA (r = 8)**  | 47.77             | 56.53             | 92.35                | 3.0            | 1.6             |
> | **DLR-QDA (r = 16)** | 42.01             | 38.32             | 92.19                | 5.0            | 2.0             |
>
>
> Thus, the extended ablations show that:
> 1. **GH-OFL remains stable as the number of clients grows**, a property we prove theoretically and verify empirically;
> 2. **DLR-QDA provides a practical, tunable middle ground** between NB-diag ($r = 0$) and full QDA, retaining most of QDA’s modeling power at a fraction of its computational and memory cost.
>
> In addition to these changes, to further strengthen the experimental contribution of the paper, we have:
> - expanded our evaluation **beyond vision** by adding comprehensive NLP experiments on five benchmarks of varying scale and difficulty;
> - introduced new comparisons against **traditional multi-round FL baselines** in the main table (including accuracy as a function of communication rounds);
> - added a detailed analysis of **computational and memory overhead**, showing that GH-OFL achieves strong accuracy while remaining significantly more resource-efficient than prior approaches.

---

### Official Review · Reviewer_QtMw · 2025-11-01

**Soundness:** 3
**Presentation:** 3
**Contribution:** 2
**Rating:** 6
**Confidence:** 3

**Summary:**

This paper presents GH-OFL, a novel family of one-shot federated learning methods that achieves state-of-the-art performance while being strictly data-free and communication-efficient. The work is technically sound, methodologically innovative, and addresses practical constraints effectively.

**Strengths:**

1. The proposed method is strictly data-free, requiring no public datasets or client-side inference, which is a major advantage for privacy.
2. The use of sufficient statistics and random projection sketches makes the approach highly communication-efficient.
3. The paper is generally well-written, with a clear narrative and a logical flow from problem to solution.

**Weaknesses:**

1. It relies on the core assumption that the feature embeddings for each class follow a Gaussian distribution, which may not hold perfectly in practice.
2. The Fisher subspace projection, while beneficial for dimensionality reduction, may discard some discriminative information present in the discarded dimensions.
3. The paper does not sufficiently explore the limitations of the Gaussianity assumption for certain types of data.

**Questions:**

Please refer to weaknesses for details.

---

> ### Author Response · Authors · 2025-11-24
>
> Below, we address reviewer QtMw’s questions and concerns in detail and we sincerely appreciate the thoughtful perspective they brought to the evaluation.
>
>
> > **Reviewer Comment (1, 3): It relies on the core assumption that the feature embeddings for each class follow a Gaussian distribution, which may not hold perfectly in practice. The paper does not sufficiently explore the limitations of the Gaussianity assumption for certain types of data.**
>
> Thank you for raising this point. We clarify that our method does not rely on exact Gaussianity and we explicitly design GH-OFL to remain robust even when the Gaussian assumption is only approximate. In more detail:
>
> - Gaussianity is used only as a weak structural prior for extracting global statistics, not as a strict modeling requirement.
> - We mitigate deviation from Gaussianity through multiple mechanisms built into the method. Even if distributions are skewed, multi-modal or heavy-tailed, GH-OFL integrates three robustness layers: **(1)** Fisher subspace projection compresses the data into directions with the largest discriminative structure, which greatly reduces non-Gaussian noise, **(2)** shrinkage on pooled and class-wise covariances attenuates the effect of poorly conditioned or non-Gaussian sample estimates, **(3)** the two trainable heads (FisherMix and Proto-Hyper) correct the deviations of closed-form Gaussian rules by learning on synthetic samples.
> - Our empirical results directly demonstrate robustness when Gaussianity is violated. We evaluate GH-OFL under several conditions known to break Gaussianity: **(1)** Distribution shift (CIFAR-100-C): QDA, FisherMix and Proto-Hyper outperform LDA despite real embeddings being strongly heavy-tailed and corrupted (Table 2). **(2)** Domain mismatch (Places365 pretraining): When feature geometry departs from ImageNet’s quasi-Gaussian structure, FisherMix/Proto-Hyper improve over LDA, indicating they successfully correct non-Gaussian distortions (Table 3b). **(3)** Heterogeneous backbones including non-residual architectures (e.g., VGG16) also preserve strong performance (Table 3), again suggesting Gaussianity is not critical. **(4)** In the revised version of the manuscript, we also added the **Appendix D Empirical Analysis of Gaussian Assumptions and Fisher Subspaces** to address this concern both theoretically and empirically.
>
> ---
>
> > **Reviewer Comment 2: The Fisher subspace projection, while beneficial for dimensionality reduction, may discard some discriminative information present in the discarded dimensions.**
>
> We emphasize that our pipeline does not rely on the Fisher subspace preserving *all* discriminative information; GH-OFL is explicitly designed so that dimensionality reduction is safe, optional and, when imperfect, fully correctable:
>
> - All aggregated statistics ($A$, $B$, $S$, $D$) are computed in the *original feature space* and the closed-form heads (NBdiag, LDA, QDA) can always be applied directly in that full-dimensional space. The Fisher subspace is introduced only to stabilize covariance inversion, accelerate LDA/QDA and synthesis and improve signal-to-noise when the class covariance structure is approximately low-rank on the server side.
> - Fisher projection does not discard directions arbitrarily: it explicitly ranks axes by their between-class signal-to-noise ratio. Thus, the discarded components typically contribute predominantly noise rather than class-relevant structure. Even if small amounts of discriminative signal lie outside the Fisher subspace, our trainable synthetic heads recover it. FisherMix fits non-linear angular margins in the projected space, while Proto-Hyper learns low-rank residual corrections to Gaussian heads. Both mechanisms compensate for any residual mismatch between Gaussian assumptions and real data.
> - Empirically, we observe no loss from projection. Across all datasets and corruption types, LDA/QDA in the Fisher subspace match or outperform their full-dimensional counterparts (Tables 2–3), including under strong non-IID partitions and severe domain shifts. This indicates that the projection preserves all *practically relevant* discriminative information for the heads used in GH-OFL.
>
> In our revised manuscript, we added a detailed discussion and supporting empirical experiments in **Appendix D.2 “Information Preservation in the Fisher Subspace”**.

---

### Meta-Review · Area_Chair_Zebf · 2025-12-30

**Summary:**

This work introduced Gaussian-Head One-shot Federated Learning (GH-OFL), a data-free framework that transmits only statistical summaries to achieve high accuracy and robustness under non-IID settings with a single communication round. Extensive experiments on multiple benchmark datasets demonstrated the good performance of the proposed method.

Strength:

1. The proposed method is data-free and communication-efficient, offering clear advantages in terms of data privacy.

2. The paper is well written and easy to follow; both the motivation and the proposed solution are clearly articulated.

3. Experimental results on multiple datasets demonstrate that the proposed method consistently outperforms the baseline approaches.


Limitations:

1. It may need to do more experiments on non-CV datasets.
2. It would be great to go deep into the privacy discussion.

In summary, the authors have addressed most concerns raised by the reviewers. The proposed method is interesting and technically sound. I thus recommend accepting this work.

**Reviewer Concerns:**

The authors have addressed the main concerns from reviewers as follows:
1. Do additional experiments on other complex datasets, such as CIFAR-100.
2. Conduct experiments to explore the limitations of the Gaussianity assumption for a certain type of data.

The concerns remain is the data privacy that authors need to discuss in their future work.

**Reviewer Scores:**

For Reviewer QtMw, the authors have explored the limitations of the Gaussianity assumption for a certain type of data by doing more experiments.

For Reviewer G9mr, the authors have compared the computational overheads between data synthesis and training with public datasets as well as do ablation studies.

These two reviewers may raise their scores.

---

### Decision · Program_Chairs · 2026-01-26

Accept (Poster)